# CRABS: A syntactic-semantic pincer strategy for bounding LLM interpretation of Python notebooks[*]

Meng Li[1], Timothy M. McPhillips[†][1], Dingmin Wang[2], Shin-Rong Tsai[1], Bertram Ludäscher[1]
[1]School of Information Sciences, University of Illinois Urbana-Champaign
[2]Department of Computer Science, University of Oxford

## Abstract

Recognizing the information flows and operations comprising data science and machine learning Python notebooks is critical for evaluating, reusing, and adapting notebooks for new tasks. Investigating a notebook via re-execution often is impractical due to the challenges of resolving data and software dependencies. While Large Language Models (LLMs) pre-trained on large codebases have demonstrated effectiveness in understanding code without running it, we observe that they fail to understand some realistic notebooks due to hallucinations and long-context challenges. To address these issues, we propose a *notebook understanding task* yielding an *information flow graph* and corresponding *cell execution dependency graph* for a notebook, and demonstrate the effectiveness of a pincer strategy that uses limited syntactic analysis to assist full comprehension of the notebook using an LLM. Our *Capture and Resolve Assisted Bounding Strategy (CRABS)* employs shallow syntactic parsing and analysis of the abstract syntax tree (AST) to *capture* the correct interpretation of a notebook between lower and upper estimates of the inter-cell I/O set—the flows of information into or out of cells via variables—then uses an LLM to *resolve* remaining ambiguities via cell-by-cell zero-shot learning, thereby identifying the true data inputs and outputs of each cell. We evaluate and demonstrate the effectiveness of our approach using an annotated dataset of 50 representative, highly up-voted Kaggle notebooks that together represent 3454 actual cell inputs and outputs. The LLM correctly resolves 1397 of 1425 (98%) ambiguities left by analyzing the syntactic structure of these notebooks. Across 50 notebooks, CRABS achieves average $F_1$ scores of 98% identifying cell-to-cell information flows and 99% identifying transitive cell execution dependencies. Moreover, 37 out of the 50 (74%) individual information flow graphs and 41 out of 50 (82%) cell execution dependency graphs match the ground truth exactly.

## 1 Introduction

Computational notebooks, such as Jupyter notebooks (Kluyver et al., 2016), provide a readable and executable medium combining code, results, and explanations that facilitates sharing of notebook functionality among users. Generally, users find it necessary to execute a notebook to discover how it works in detail, confirm that it works correctly, and assess its potential for reuse. However, only one in four[1] notebooks discovered on GitHub that specify both Python version and execution order can be re-executed without errors (Pimentel et al., 2019). Most of these errors stem from module dependency and data accessibility

---

[*]Code and data available at `https://github.com/cirss/crabs/tree/v1.0.0`.

[†]Corresponding author: tmcphill@illinois.edu

[1]Among 1,081,702 unique Python notebooks discovered on GitHub, Pimentel et al. (2019) attempted to execute the 863,878 notebooks for which both Python version and execution order could be determined, and found that only 24% of the latter notebooks could be executed without errors using either the declared dependencies or a comprehensive, preconfigured Anaconda environment.

issues that are often difficult or even impossible to resolve. For this reason, facilitating interpretability of notebooks in the absence of re-exeution would represent a major step toward improving reproducibility and lowering barriers to reuse. Such interpretability requires understanding what individual variables mean, where they are defined, and where they are subsequently used. Answers to such questions are critical for evaluating, reusing, and adapting a notebook for new tasks.

Workflow management systems and frameworks (e.g., Galaxy (Goecks et al., 2010), Kepler (Ludäscher et al., 2006), Taverna (Oinn et al., 2004), RestFlow (Tsai et al., 2013), and Snakemake (Köster & Rahmann, 2012)) make these dataflows explicit, thereby facilitating evaluation, reuse, and repurposing. However, these systems generally require that code be structured or restructured to fit the workflow model each implements. In contrast, YesWorkflow (McPhillips et al., 2015) provides means to reveal the dataflows implicit in scripts without requiring users to modify their code. Instead, the author of the code declares the dataflows via annotations included in comments. The YesWorkflow annotation vocabulary models scripts as information flow graphs, i.e., collections of *program blocks* connected by *channels* representing paths of data flow. When applied to Python notebooks, code cells naturally correspond to the YesWorkflow program blocks, while variables assigned in one cell and used in another correspond to channels. Viewing a notebook this way suggests that it should be feasible to generate meaningful information flow graphs for notebooks automatically, thereby making it clear what a notebook does and how it works—without running it.

Extracting the information flow graph from a computational notebook requires not just identifying how data flows between code cells, but also how user-defined functions and classes declared in one cell are used another. Fortunately, recognizing code cells that define or call user-defined functions and classes is relatively straightforward via syntactic parsing and analysis of the abstract syntax tree (AST). In contrast, identifying which cells actually assign new values to variables or update them is challenging without comprehensively analyzing the source code of imported modules or executing the cells. For example, the statement `phone_data.dropna(inplace=True)` updates the `phone_data` object in place, while the statement `phone_data.truncate(before=1, after=3)` returns information from the `phone_data` object without modifying it. Without understanding the mechanisms of these two methods, or comparing the `phone_data` object before and after executing the code, it is not possible to determine whether the `phone_data` object is updated or not. Therefore, how data flows between code cells can be ambiguous if only the syntactic structure apparent in code cells is taken into account.

We observe that experienced data scientists often can resolve these ambiguities without accessing source code for imported modules and without executing the cells by hand. This raises the question: can Large Language Models (LLMs) do the same thing? LLMs pretrained on large codebases have been shown to be effective on code-related tasks, e.g., bug detection (Wang et al., 2024), code understanding (Nam et al., 2024), code completion (Zhang et al., 2023; Cheng et al., 2024), code generation (Yin et al., 2023), and code summarization (Haldar & Hockenmaier, 2024). For these reasons, we hypothesize that LLMs can help resolve ambiguites left by purely analyzing the syntactic structure of the notebook itself, i.e., just the code in its cells.

At present, however, LLMs cannot reliably analyze entire notebooks on their own. As we show in Section 5.3, LLMs face hallucination issues and long-context challenges with the result that they sometimes identify variables that do not exist, and for longer notebooks frequently return information flow graphs with the wrong number of code cells. To address these issues, we adopt an approach that reduces the overall workload assigned to the LLM and limits its tasks to those that entail semantically informed analysis of the code.

In this paper, we propose the Capture and Resolve Assisted Bounding Strategy (CRABS), a syntactic-semantic hybrid strategy for extracting the information flow graph from a Python notebook. Figure 1 is an overview of the steps employed by CRABS to generate an information flow graph for a given Python notebook. CRABS depends critically on the concept of a notebook's *inter-cell I/O set*, defined as the set of all flows of information into or out of individual cells via variables. We first produce lower and upper estimates of the

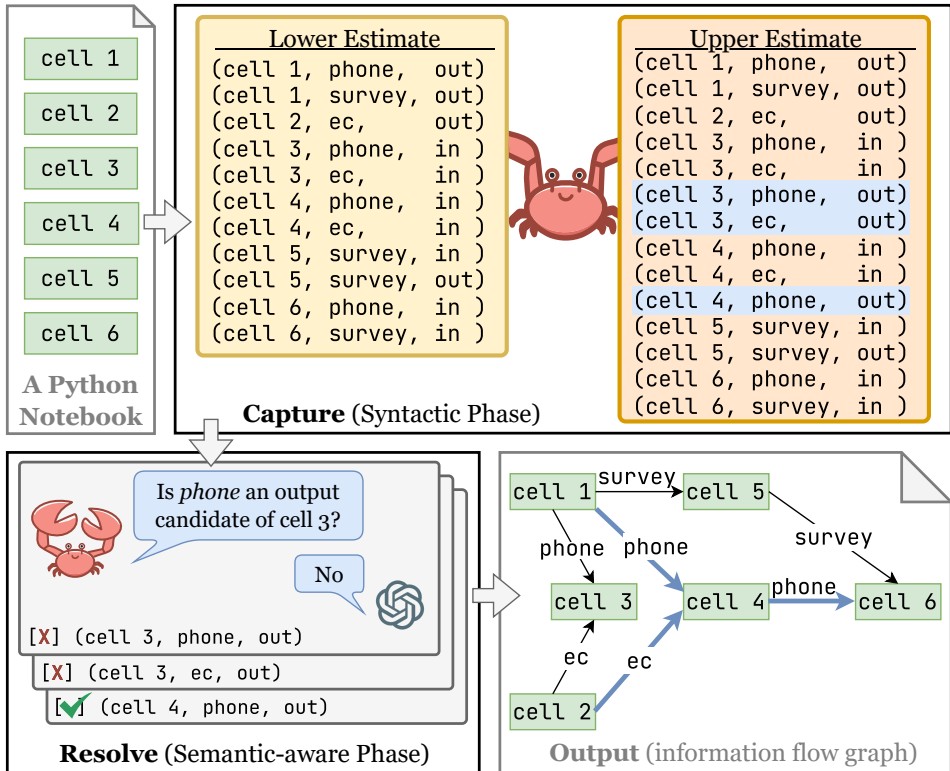

Figure 1: An overview of CRABS using the example notebook in Figure 2. CRABS first produces lower and upper estimates of the inter-cell I/O set in a syntactic phase, then resolves the differences between these estimates using an LLM in a semantic-aware phase, and finally constructs an information flow graph with thick blue edges indicating this resolved set.

inter-cell I/O set based on the syntactic structure of the notebook, then prompt an LLM to resolve the differences between these two estimates (i.e., ambiguities) one by one. To evaluate this approach, we apply CRABS to 50 representative, highly up-voted Python notebooks from the Meta Kaggle Code dataset (Plotts & Risdal, 2023).

Our **main contributions** are: (1) annotating 50 notebooks, manually identifying the 2437 cell-to-cell information flows and 7998 transitive cell execution dependencies in these notebooks to serve as the ground truth for our analyses; (2) producing lower and upper estimates of inter-cell I/O sets that bound the ground truth represented by our notebooks; (3) demonstrating that LLMs correctly resolve individual ambiguities in how data flows between code cells in these notebooks with 98% accuracy; (4) showing that the CRABS strategy of prompting LLMs to answer simpler, highly specific questions about a notebook mitigates long-context challenges and eliminates hallucination of variables; and (5) generally demonstrating the effectiveness of a pincer strategy that places bounds both on the total workload assigned to the LLM, and on the individual responses that it can return.

## 2   Related Work

**Information flows in computational notebooks**. Existing approaches for revealing information flows in computational notebooks depend on human annotations (McPhillips et al., 2015; Ramasamy et al., 2023), structuring code to follow a particular set of guidelines (Carvalho et al., 2017; Koop & Patel, 2017), or using runtime information (Koop & Patel, 2017; Wenskovitch et al., 2019; Brachmann et al., 2020; Li et al., 2024).

```
# cell 1
import pandas as pd
phone = pd.read_csv("phone_records.csv")
survey = pd.read_csv("user_surveys.csv")
```

```
# cell 2
# exclusion criterion
ec = "validhourscount < 12"
```

```
# cell 3
len(phone.query(ec)) / len(phone)
```

```
# cell 4
phone.drop(
    phone.query(ec).index,
    inplace=True)
```

```
# cell 5
symptoms = ["pain", "fatigue", "anxiety"]
scores = survey[symptoms].mean(axis=1)
survey["score"] = scores
```

```
# cell 6
data = phone.merge(survey, on="date")
```

Figure 2: An example Python notebook with six code cells, representing a simplified data preparation pipeline for symptom burden prediction using smartphone records.

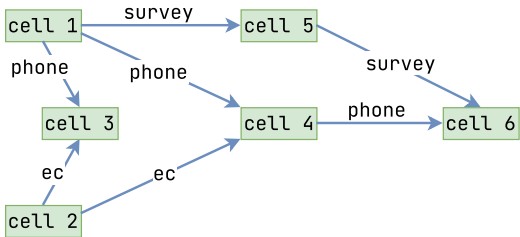

Figure 3: Ground-truth information flow graph for the example notebook (Figure 2).

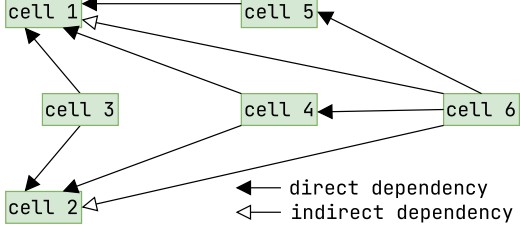

Figure 4: Ground-truth cell execution dependency graph for the example notebook (Figure 2).

**LLMs and computational notebooks**. Pre-trained language models have been leveraged to generate feedback on answers to open-ended questions about instructional coding or math notebooks (Matelsky et al., 2023); and fine-tuned for code generation (Huang et al., 2022; Yin et al., 2023; Li et al., 2023; Huang et al., 2024). Cao et al. (2024) explore multimodal agents for data science notebook generation.

**Dataflow, LLMs, and code-related tasks**. Dataflow analysis is fundamental to compiler design. Guo et al. (2021) leverage dataflow structure of code to train a language model for code-related tasks. Wang et al. (2024) employ LLMs to generate customized AST-based parsers for dataflow analysis in support of bug detection. Cheng et al. (2024) employ an AST-based parser and dataflow analysis to construct a repo-specific context graph for repository-level code completion.

## 3   A Model of Information Flow in Python Notebooks

CRABS views a Python notebook as a sequence of $n$ code cells $C = (c_1, c_2, ..., c_n)$ where $c_j$ is the $j$th code cell (markdown cells are ignored). CRABS further assumes that cells are executed in order, i.e., from the top to the bottom of the notebook. This assumption limits the possible interpretations of information flows in a notebook; i.e., a value assigned to a variable in cell $c_j$ cannot be used in cell $c_{j-1}$ or earlier. We represent the $m$ distinct units of information (i.e., variable values and code references) that flow between cells as $I = \{i_1, i_2, ..., i_m\}$ and define an *information flow graph* as a set of information flows $F \subseteq \{(c_s, c_t, i_k) \mid 1 \leq s < t \leq n\}$, where $i_k$ flows from source cell $c_s$ to target cell $c_t$. The information flow graph in turn implies a *cell execution dependency graph*, i.e., the set of transitive dependencies $D \subseteq \{(c_t, c_s) \mid 1 \leq s < t \leq n\}$, where cell $c_t$ directly or indirectly depends on cell $c_s$. The cell execution dependency graph can be derived from the information flow graph by (1) recognizing a direct dependency between two cells if at least one information flow exists between them in the information flow graph; and (2) computing the transitive closure of this graph to infer indirect dependencies.

The model of information flow that CRABS adopts for Python notebooks aims to support a number of crucial use cases. Using the information flow graph, it is straightforward

to answer questions such as "Which cells may have contributed to the final value of a particular variable?", "Which cells may have been influenced by a particular variable?", and "Which cell defines a particular function or class and which cells call it?" Similarly, using the cell execution dependency graph, one can answer questions such as "What cells depend (directly or indirectly) on a particular cell?" and "What cells influence a particular cell?"

As an example, consider the simple, six-cell data preparation notebook represented in Figure 2 and the corresponding information flow and cell execution dependency graphs in Figure 3 and Figure 4. The information flows $(c_1, c_5, \text{survey})$ and $(c_5, c_6, \text{survey})$ can easily provide cell 1 and cell 5 as the answer to "What cells may have contributed to the value of the survey dataframe received by cell 6?" The transitive dependency $(c_6, c_5)$ can easily provide cell 6 as the answer to "What cells depend on cell 5?"

We refer to the problem of identifying the information flow graph and corresponding cell execution dependency graph for a notebook as the *notebook understanding task*. In the remainder of this paper, we describe and evaluate how CRABS performs this task.

## 4 Elements of CRABS

Constructing the information flow graph and corresponding cell execution dependency graph for a notebook depends crucially on determining its inter-cell I/O set. The approach taken by CRABS to determine the inter-cell I/O set is based on the observation that some members of the set are certain by shallow syntactic parsing and analysis of the AST, while others are uncertain but may be resolved by taking the semantic meaning of these code cells into consideration. CRABS therefore performs the notebook understanding task in two phases: a syntactic phase and a semantic-aware phase. The syntactic phase employs shallow syntactic parsing to extract an AST based purely on the notebook itself, and analyzes the extracted AST to yield lower and upper estimates of the inter-cell I/O set. The semantic-aware phase resolves the ambiguities between these lower and upper estimates using an LLM.

We represent the inter-cell I/O set as $S \subset \{(c, i_k, tag) \mid c \in \{c_s, c_t\}, tag \in \{\text{out}, \text{in}\}\}$, where the out *tag* denotes that information $i_k$ flows out of source cell $c_s$, and the in *tag* denotes that information $i_k$ flows into target cell $c_t$. Therefore, it is critical to identify four items for each cell, i.e., data inputs, data outputs, code declarations, and code references. If data $i_k$ is defined in cell $c_s$ and used in cell $c_t$, then $i_k$ is a *data output* of $c_s$ and a *data input* of $c_t$. However, when cell $c_s$ is examined in isolation, we cannot determine whether there is in fact a subsequent cell $c_t$ that uses $i_k$. Because we do not provide any information about subsequent cells when asking the LLM to resolve ambiguous information flows associated with a particular cell, we refer to *data output candidates* rather than *data outputs* in these prompts. Code references, e.g., functions and classes declared in cells and used in later cells, are treated analogously to data. In the rest of this section, we describe the syntactic phase and semantic-aware phase based on these four items. Relevant examples are attached in the Appendix.

### 4.1 Syntactic Phase

The syntactic phase of CRABS performs a limited static analysis of the code in a notebook to yield two estimates of the inter-cell I/O set, representing lower and upper estimates of the actual flows of information into or out of each cell. It produces two estimates because CRABS aims to interpret Python notebooks without accessing the source code for any dependencies, e.g., code defined outside the notebook and accessed via function calls. Deeper or more sophisticated syntactic parsing and analysis of the AST that descends into each function called from a cell likely would achieve tighter lower and upper estimates of the inter-cell I/O set, leaving fewer or even no ambiguities to be resolved by an LLM. However, because we restrict our syntactic analysis to code explicitly included in a notebook's cells, CRABS does not need to resolve software dependencies. Moreover, even comprehensive static analysis will not yield a unique prediction of the dataflow between cells when conditional control

```python
import pandas as pd
train = pd.read_csv("train.csv")
test = pd.read_csv("test.csv")
datasets = [train, test]
```

```python
train.dropna(inplace=True)
```

Figure 5: An example shared reference issue.

flow structures are employed, e.g., if variables are assigned, used, or updated conditionally (see Appendix A.1.1).

CRABS makes three basic assumptions about the code in Python notebooks relevant to the syntactic phase. First, execution flow follows a strict top-to-bottom order, starting from an initial state with all variables, imports, and other runtime information cleared from memory. Second, all code is reachable. And third, function and class names are not reused as variable names or vice versa (see Appendix A.1.2). In addition, our current implementation has four further limitations. Global variables that are accessed or modified within a function, without being explicitly passed as arguments, will not be recognized on this basis as data inputs or data output candidates by our system (see Appendix A.1.3). Lines of code employing languages other than Python, e.g., shell commands invoked using the *!* syntax, or *IPython* magic commands invoked using the % syntax are ignored (see Appendix A.1.4). A third limitation related to shared data references (aliased variables) is more complex. This issue arises when multiple variables reference the same mutable object in memory, and the object is modified. For example, the first code cell in Figure 5 defines two *Pandas* DataFrames, `train` and `test`. The `datasets` object is a list that references two mutable objects, `train` and `test`. In the second code cell, `train` is modified in place, resulting in an implicit change in `datasets`. We define this phenomenon as hidden modifications, wherein a variable name (i.e., `datasets`) is not explicitly written in a code cell but should still be considered a data output candidate. The current implementation of CRABS detects only those shared references explicitly indicated by assignment (e.g., `x = y`) or by inclusion in collections (i.e., mutable objects stored in a list, a tuple, or a dictionary). This covers all instances of shared references in our dataset; we will extend CRABS to detect shared references established by other mechanisms in the future. Finally, CRABS produces a single information flow graph per notebook, even when multiple execution paths are possible (e.g., due to conditional statements) given different input data. When different execution paths yield different information flows, the output of CRABS represents the union of those flows.

The *lower estimate* computed by the syntactic phase represents the set of cell inputs and output candidates considered certain (i.e., unambiguous) given just the code apparent in the cell. This lower estimate is computed using the following rules: (1) Variables explicitly passed to a function are regarded as data inputs only. (2) The iterator of a *for* loop, a *while* loop, or a comprehension is regarded as a data input only, unless it is directly reassigned in the loop body. (3) Because CRABS yields information flows for all possible execution paths, it generally treats code in every branch of a conditional statement as contributing to the inputs and the output candidates of the cell. (4) However, when a conditional statement occurs inside a loop, the *intersection* of all possible execution orders is considered (see Appendix A.1.5 for an illustrative example). (5) The effect of shared references (e.g., aliased variables) is not considered. Figure 6a and Figure 6c show the information flow graph and the corresponding cell execution dependency graph generated from the lower estimate.

The *upper estimate* represents the set of cell inputs and output candidates that are either certain or ambiguous, which is a superset of both the ground truth and the lower estimate. This upper estimate is computed using the following rules: (1) Variables explicitly passed to a function are regarded as both data inputs and data output candidates. (2) The iterable of (e.g., the collection traversed by) a *for* loop, a *while* loop, or a comprehension is regarded both as a data input and a data output candidate. (3) As with the lower estimate, code on all branches of a conditional statement are generally considered to contribute to the inputs and the output candidates of the cell. (4) But unlike the lower estimate, when a conditional statement occurs inside a loop the *union* of all possible execution orders is considered.

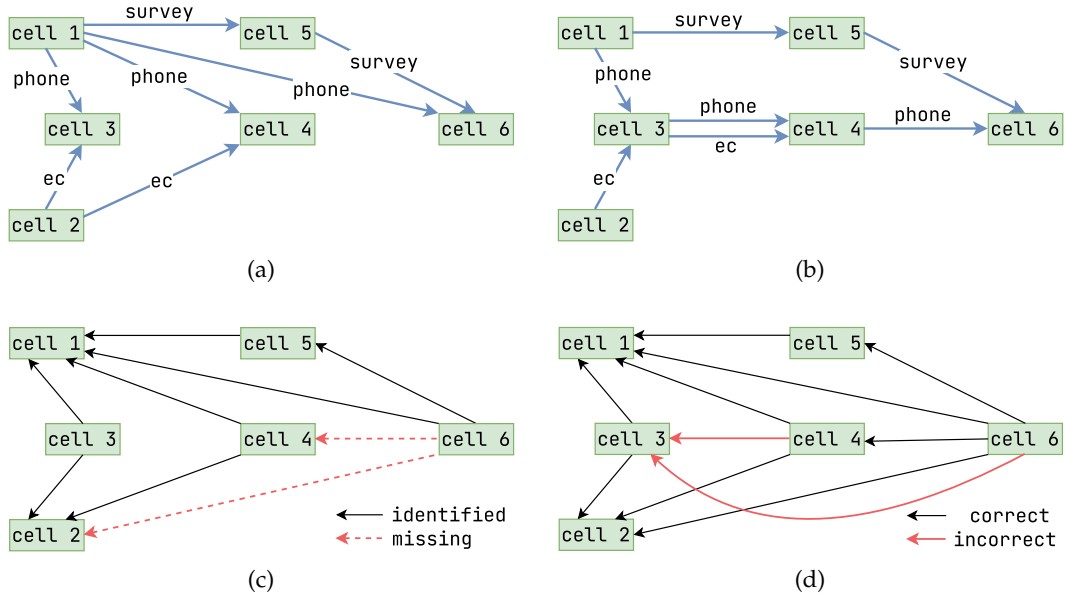

Figure 6: Information flow graphs derived from the (a) *lower* and (b) *upper* estimates of the inter-cell I/O set for the notebook in Figure 2; (c) transitive dependencies (solid black) extracted from Figure 6a augmented with missing edges (dotted red); and (d) transitive dependencies extracted from Figure 6b with correct (black) and incorrect (red) dependencies.

(5) The effect of shared references is taken into account. Figure 6b and Figure 6d show the information flow graph and the corresponding cell execution dependency graph generated from the upper estimate.

Figures 6c and 6d demonstrate the subgraph and supergraph relations between the cell execution dependency graphs generated from lower and upper estimates, respectively, and the ground truth. The identified dependencies (black) in Figure 6c represent a subgraph of the ground truth, whereas the correct (black) and incorrect (red) edges in Figure 6d together represent a supergraph of the ground-truth cell execution dependency graph.

### 4.2 Semantic-aware Phase

We hypothesize that pre-trained language models can help resolve ambiguous members of the inter-cell I/O set identified by the syntactic phase. We resolve two different kinds of ambiguities, (1) checking if data is an input of a given code cell and (2) checking if data is an output candidate of a given code cell. An LLM resolves these ambiguities cell by cell using a zero-shot in-context learning approach. Example prompts are included in Figure 15 and Figure 16 of the Appendix. Requesting binary answers (i.e., "yes" or "no") eliminates hallucinations of variables; all data inputs and output candidates are selected from the given syntactic phase results. Prompts include information about shared references defined in previous cells only when shared references are among the ambiguous data. Since the names of shared references are identified in the syntactic phase, the LLM can be prompted for each code cell independently, improving time efficiency.

## 5 Experiments

### 5.1 Dataset

Our dataset comprises a set of 50 annotated Python notebooks. We assembled the dataset from the 50 most up-voted Python notebooks in Kaggle that met the following selection criteria:

| per notebook | sum | avg | min | 25% | 50% | 75% | max |
|---|---|---|---|---|---|---|---|
| code cells | 1493 | 30 | 2 | 20 | 25 | 40 | 76 |
| lines | 10576 | 212 | 26 | 34 | 125 | 288 | 1111 |
| information flows | 2437 | 49 | 4 | 23 | 32 | 78 | 153 |
| transitive dependencies | 7998 | 160 | 1 | 55 | 75 | 157 | 922 |

Table 1: Summary statistics for the CRABS dataset.

1. Notebook implements a machine learning workflow. Notebooks containing basic tutorials and other instructional materials were excluded.

2. Variable use is explicit and clear. We excluded Python notebooks that used global variables within functions, or reused variable names as function or class names (and vice versa).

3. Information flows comprise a single connected graph. Notebooks excluded on this basis either implemented multiple independent workflows, or routed cell-to-cell dataflows through external files.

We determined which notebooks met these criteria by carefully inspecting their contents. We inspected the 104 most up-voted Python notebooks to find 50 meeting these criteria. Reasons for excluding the other 54 notebooks are indicated in Table 5 of the Appendix. For each notebook we selected, we extracted the corresponding kernel version ID from the Meta Kaggle dataset (Risdal & Bozsolik, 2022), and retrieved the notebook from the Meta Kaggle Code dataset (Plotts & Risdal, 2023).

We annotated each of these notebooks by hand, identifying the inputs and outputs of each cell. These annotations serve as the ground truth for evaluating CRABS. Table 1 summarizes the characteristics of the selected 50 notebooks. These notebooks generally are nontrivial, on average containing 49 information flows and 160 transitive dependencies each. Detailed statistics for each notebook are reported in Table 6 of the Appendix. As these are highly up-voted Kaggle notebooks representing both high-quality tutorials and real-world notebooks for machine learning pipelines, we consider them representative of general data science and machine learning notebooks.

## 5.2 Evaluation Metrics

We evaluate how well CRABS performs the notebook understanding task at three different levels. The *Exact Match* (*EM*) metric captures the notebook-level performance, and is defined as the percentage of notebooks for which the generated information flow graph or cell execution dependency graph exactly matches the ground truth. At the next level of detail, we evaluate how well CRABS predicts inter-cell information flows and execution dependencies using the metrics *Precision*, *recall*, $F_1$ score, and *accuracy*. These metrics are widely used to evaluate information retrieval (Kobayashi & Takeda, 2000), classification (Hossin & Sulaiman, 2015), and link prediction (Daud et al., 2020) tasks, and are easily adapted to our task by treating the returned information flows and transitive dependencies as predicted links, and the human annotated flows as ground truth. Finally at the level of resolving individual syntactic ambiguities, we evaluate the *accuracy* of LLM responses defined as the fraction of questions answered correctly.

## 5.3 Main Results

Results were obtained using the `gpt-4o-2024-08-06` version of the GPT-4o model (OpenAI, 2024a) with `temperature` set to 0. We compared the performance of CRABS against a baseline approach which employs zero-shot prompting of an LLM for the whole notebook with carefully designed chain-of-thought prompts (Wei et al., 2022). An example prompt is shown (in part) in Figure 14 of the Appendix. We observe that the LLM completely fails to understand 10 out of 50 (20%) notebooks, returning the wrong number of cells and earning zero values for all metrics. These 10 notebooks are long with 52 cells on average, where the

| Method | Information Flows (%) | | | | | Transitive Dependencies (%) | | | | |
|---|---|---|---|---|---|---|---|---|---|---|
| | Prec. | Rec. | $F_1$ | Acc. | EM | Prec. | Rec. | $F_1$ | Acc. | EM |
| baseline | 73.51 | 67.89 | 70.10 | 64.39 | 26 | 79.45 | 68.95 | 72.90 | 68.59 | 36 |
| *upper estimate* | 56.31 | 56.39 | 56.35 | 42.70 | 8 | 53.06 | **100** | 66.08 | 53.06 | 8 |
| *lower estimate* | 92.05 | 89.91 | 90.79 | 85.19 | 42 | **100** | 86.65 | 91.98 | 86.65 | 42 |
| CRABS | **98.61** | **98.53** | **98.57** | **97.32** | **74** | 99.42 | 99.95 | **99.67** | **99.37** | **82** |

Table 2: Main results for the notebook understanding task.

LLM faces long-context challenges. For other notebooks, the LLM sometimes hallucinates, describing variables not present at all in the reported cells.

Table 2 summarizes the performance of CRABS for the notebook understanding task. These results are average scores across 50 notebooks in our dataset. We observe a percentage-point increase of 28 in $F_1$ score, 32 in accuracy, and 48 in EM for information flow graphs; and percentage-point increases of 26 in $F_1$ score, 30 in accuracy, and 46 in EM for cell execution dependency graphs. These results demonstrate the effectiveness of CRABS for the notebook understanding task. We include in this table metrics for the *lower estimate* and *upper estimate* components of CRABS for comparison with the full CRABS approach. For the transitive dependencies, the precision for *lower estimate* and the recall for *upper estimate* are each 100%, consistent with the subgraph and supergraph relations described in Section 4.1. We note that in contrast to the baseline, CRABS yields plausible outputs (i.e., earns non-zero contributions for each metric) for all notebooks; we attribute this improvement to the cell-by-cell prompting strategy CRABS employs which avoids the long-context challenges faced by the baseline method.

CRABS yields information flow graphs that exactly match the ground truth for 37 notebooks. Of these, 4 return an identical lower and upper estimate. As an LLM resolves ambiguities between these estimates, the LLM is not used for these 4 notebooks. The metrics in Table 2 thus do not fully reflect the capability of the LLM for resolving ambiguities. We evaluate such capability at a finer level of granularity, finding that 1397 out of 1425 (98%) ambiguities are correctly resolved by the LLM.

## 5.4 Ablation Studies

Because cell execution dependency graphs are extracted from information flow graphs, we only discuss the information flow graphs in this section. For completeness, the results of the ablation studies with respect to cell execution dependency graphs are reported in Table 8 and Table 9 of the Appendix. Detailed results for each notebook are reported in Table 10 of the Appendix.

**S1. Effectiveness of the syntactic phase**. Instead of using the syntactic-semantic hybrid approach, we test the performance of an approach without the syntactic phase, asking the LLM to identify cell inputs and output candidates without providing any hints from the syntactic processor. An example prompt is in Figure 17 of the Appendix. Table 3 shows percentage-point decreases of 7 in $F_1$ score, 10 in accuracy, and 36 in EM, demonstrating the effectiveness of syntactic parsing.

**S2. Effectiveness of cell-by-cell prompting**. Instead of using the cell-by-cell prompting approach, we test the performance of using a single prompt for the whole notebook. Part of the example prompt is shown in Figure 18 of the Appendix. Table 3 shows percentage-point decreases of 16 in $F_1$ score, 20 in accuracy, and 38 in EM. The main reason it drops dramatically is the long-context challenge, which causes the LLM to fail to handle multiple code cells using a single prompt. Five notebooks, each containing an average of 54 code cells, return the wrong number of cells. Detailed metrics for the remaining 45 notebooks are shown in Table 7 of the Appendix. These results demonstrate that cell-by-cell prompting outperforms the single prompt approach.

| Method | Information Flows | | | | |
|---|---|---|---|---|---|
| | Prec.(%) | Rec.(%) | $F_1$(%) | Acc.(%) | EM(%) |
| CRABS | **98.61** | **98.53** | **98.57** | **97.32** | **74** |
| w/o syntactic phase | 95.29 | 89.38 | 91.22 | 86.37 | 38 |
| w/o cell-by-cell prompting | 82.97 | 80.93 | 81.71 | 76.85 | 36 |

Table 3: Ablation study results.

## 6  Conclusion

We have demonstrated that the notebook understanding task is feasible even in the absence of comprehensive code analysis and without executing notebook cells. CRABS achieves this by employing simple syntactic analysis to limit the tasks given to an LLM and to delimit its possible answers. This success suggests that such a pincer strategy may represent a promising general approach to combining neural and symbolic methods.

## 7  Future Work

Work is underway to enhance CRABS to extract and represent multiple distinct information flow graphs for a single notebook. This refinement will enable more precise and comprehensive analysis by capturing multiple alternative information flows that may emerge under different conditions.

We are currently investigating approaches for optimizing the overall execution performance of CRABS, including reducing the total latency of the notebook understanding task. Preliminary analysis shows that a sequential implementation of CRABS achieves lower latency for this task than the baseline for notebooks with a small number of ambiguities, but that this latency increases beyond that of the baseline approach as the number of ambiguities grows. Further experiments demonstrate that a concurrent implementation of CRABS consistently outperforms the baseline in terms of latency, regardless of the number of ambiguities. These findings (see Appendix A.2) point to the potential for CRABS to achieve the performance required for practical, interactive applications. Smaller LLMs may offer further improvements in execution speed; see Appendix A.3 for early results evaluating CRABS on such models.

Finally, in addition to serving as an effective neuro-symbolic technique for notebook understanding, we view CRABS as the foundation for an experimental platform for investigating the information used, the skills employed, and the sub-problems that must be solved for any intelligence, artificial or natural, to understand a Python notebook via its information flow graph and the corresponding cell execution dependency graph.

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

# A    Appendix

## A.1    Illustrative Python notebooks

### A.1.1    Notebook with information flow dependent on input data

The notebook illustrated in Figure 7 exhibits different patterns of information flow depending on the input dataset. Because line 3 in cell 2 will be executed only when the phone_records.csv file has more than 100 rows, information flows from cell 2 to cell 3 only for runs on such input files.

```
1   # cell 1
2   import pandas as pd
3   phone_data = pd.read_csv("phone_records.csv")

1   # cell 2
2   if phone_data.shape[0] > 100:
3     phone_data = phone_data.sample(n=100)

1   # cell 3
2   phone_data.head()
```

Figure 7

### A.1.2    Notebook with clashing variable and function names

Figure 8 shows an example case that violates the assumption of distinct data and code names. Cell 1 defines a variable named add_one, cell 2 reuses the variable name as a function name, and cell 3 calls the function. Our system cannot identify that the add_one variable is a data output candidate of cell 1 and a data input of cell 2, resulting in a further failure to extract the data flow (cell 1, cell 2, add_one) and the cell execution dependency (cell 2, cell 1).

```
1   # cell 1
2   add_one = 1

1   # cell 2
2   add_one = add_one + 1
3   # Reassign the variable name to a function
4   def add_one(v):
5       return v + 1

1   # cell 3
2   add_one(5)
```

Figure 8

### A.1.3    Global variable accessed in a function

Cell 1 of the notebook in Figure 9 defines a global variable data_file_path and cell 2 accesses it directly from the load_data function. Our system cannot identify that data_file_path is a data input of cell 2, resulting in a further failure to recognize the information flow (cell 1, cell 2, data_file_path) and the cell execution dependency (cell 2, cell 1).

```
1   # cell 1
2   data_file_path = "data.csv"
```

```
1   # cell 2
2   import pandas as pd
3   def load_data():
4       data = pd.read_csv(data_file_path)
5       return data
6   data = load_data()
```

Figure 9

### A.1.4 Notebook with non-Python statements

Cell 1 in Figure 10 uses the `%%capture` magic command to capture the standard output of the code cell, storing it as a variable named `captured_stdout` that is later used by cell 2. Our system skips cell 1 due to the magic command, and fails to extract the information flow (`cell 1, cell 2, captured_stdout`) and the cell execution dependency (`cell 2, cell 1`).

```
1   # cell 1
2   %%capture captured_stdout
3   out = 1
4   print(out + 1)
```

```
1   # cell 2
2   captured_stdout.show()
```

Figure 10

### A.1.5 Conditional statements in a loop

Cell 1 in Figure 11 defines count and cell 2 has an *if* statement inside a loop, assigning 0 to count on the first iteration and incrementing count by one during each of the later two iterations. While the *lower estimate* considers the intersection of all possible branch execution orders, finding cell 1 and cell 2 to be independent; the *upper estimate* considers the union of all possible branch execution orders, finding the information flow (`cell1, cell2, count`) and a cell execution dependency (`cell 2, cell 1`).

```
1   # cell 1
2   count = 1
```

```
1   # cell 2
2   for i in range(3):
3       if i == 0:
4           count = 0
5       else:
6           count = count + 1
```

Figure 11

## A.2 Execution time analysis

Figure 12 shows that, when using the sequential implementation of CRABS, the execution time for a notebook, defined as the total of the latencies of the individual requests involved in analyzing the notebook, is proportional to the total number of ambiguities. The ordinary least squares (OLS) linear regression yields a slope of 0.94 seconds per ambiguity and an $R^2$ of 0.96. In contrast, the OLS fit of the baseline execution time against the number of

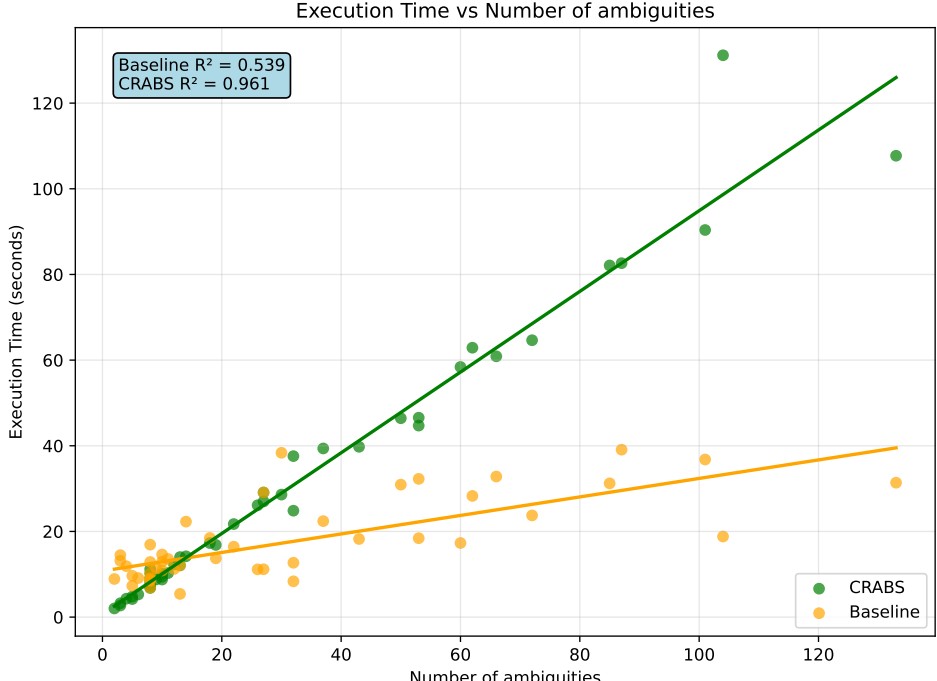

Figure 12: Scatter plot of execution time vs. number of ambiguities per notebook with least squares lines.

ambiguities is substantially weaker with an $R^2$ of 0.54, though its slope of 0.22 seconds per ambiguity remains informative. The intersection point of these two regression lines supports the observation that CRABS is generally up to 5 times faster than the baseline for notebooks with fewer than 14 ambiguities, and as much as 7 times slower than the baseline when there are more ambiguities. A separate regression against the total number of information flows in a notebook (Figure 13) better predicts the baseline execution time with an $R^2$ of 0.76. For the 4 notebooks containing no ambiguities (i.e., for each cell in these notebooks the parser-generated lower and upper estimate of the inter-cell I/O sets are identical), CRABS is approximately 3 orders of magnitude faster than the baseline because no prompting of an LLM is necessary; these 4 notebooks were excluded from the linear regressions discussed above.

While the sequential implementation of CRABS resolves exactly one ambiguity per request in a sequential manner, the concurrent implementation sends requests to an LLM without waiting for prior responses and achieves performance improvements ranging from 4 to 24 times faster than the baseline on a per-notebook basis, with per-notebook execution times ranging from 0.6 to 8.3 seconds.

## A.3 Effectiveness of CRABS on LLMs other than GPT-4o

Table 4 validates the effectiveness of CRABS on smaller models, including open- and closed-source, general-purpose and code-focused language models—GPT-4o mini (OpenAI, 2024b), Qwen3-8B (Yang et al., 2025), and Qwen2.5-Coder-7B-Instruct (Hui et al., 2024). We observe that smaller models face greater long-context challenges. For example, the GPT-4o mini model fails to understand 25 notebooks under the baseline setting and 20 notebooks without cell-by-cell prompting, returning the wrong number of cells and resulting in zero values for all metrics in each of these notebooks.

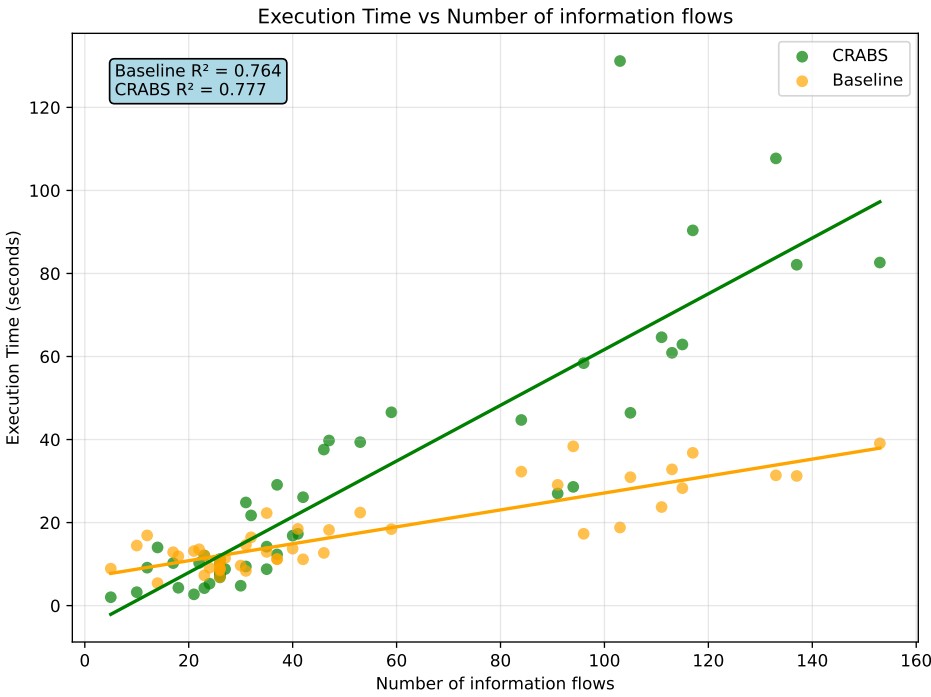

Figure 13: Scatter plot of execution time vs. number of information flows per notebook with least squares lines.

| LLM | Method | Information Flows (%) | | | | |
|-----|--------|------|------|-------|------|-----|
| | | Prec. | Rec. | $F_1$ | Acc. | EM |
| GPT-4o-mini | baseline | 43.19 | 40.65 | 41.53 | 37.40 | 12 |
| (2024-07-18) | CRABS | **83.01** | **82.87** | **82.94** | **74.85** | **38** |
| | w/o syntactic phase | 71.58 | 65.16 | 67.73 | 55.09 | 10 |
| | w/o cell-by-cell prompting | 54.69 | 53.57 | 54.01 | 50.55 | 28 |
| Qwen3-8B | baseline | 45.75 | 38.92 | 41.36 | 34.77 | 6 |
| | CRABS | 75.04 | **73.75** | **74.31** | **63.48** | **22** |
| | w/o syntactic phase | **87.56** | 45.77 | 57.21 | 43.78 | 8 |
| | w/o cell-by-cell prompting | 37.07 | 36.86 | 36.95 | 32.29 | 16 |
| Qwen2.5-Coder | baseline | 19.47 | 13.87 | 15.80 | 12.64 | 2 |
| (7B-Instruct) | CRABS | **77.13** | **76.01** | **76.52** | **66.21** | **26** |
| | w/o syntactic phase | 64.90 | 56.64 | 59.81 | 47.24 | 6 |
| | w/o cell-by-cell prompting | 35.55 | 33.49 | 34.36 | 31.71 | 16 |

Table 4: CRABS results for LLMs other than GPT-4o.

## A.4 Additional figures and tables

```
Given a Python Jupyter Notebook, please extract input and output variables, custom
defined code (including functions and classes), referred code (including functions
and classes) of each code cell and return a JSON string which is a list of
dictionaries. Each dictionary has "inputs", "outputs", "defines_code", and
"refers_code" keys. Each dictionary is a code cell in the notebook. All modules,
builtin function names should be ignored. The order of these dictionaries in the
list should be the same as the order of the code cells in the notebook.

Here are the steps:

Step 1: extract the inputs and output candidates of each code cell. An input of a
cell is a variable that is used in the cell. Usually it is not defined in current
cell, but in some previous cells. An exception could be a variable that is defined
in the current cell but used before it is defined. An output candidate of a cell is
a variable that is defined, updated, or mutated in the cell.

Important cases for output candidates:
1. Method Calls (object.method()) or Function Calls (function(object))
- If a method (or a function) modifies the object in place (e.g., updating data,
changing the structure, or adjusting properties in place), the object is an output
candidate.
- If a method (or a function) does not modify the object (e.g., retrieving data,
describing data, visualizing data, or accessing properties), the object is not an
output candidate.
- If a method (or a function) returns a new object (e.g., creating a copy) without
modifying the original, the original object is not an output candidate.
2. Reassignment (variable = ...)
If a variable is assigned a new value, it is an output candidate.
3. Shared References (Aliased Variables)
- If multiple variables reference the same mutable object (e.g., through assignment
or being stored inside a data structure), modifying the object in place through one
reference makes all references to that object output candidates.
- Modifying the container in place makes only the container an output candidate,
but not its elements.
- Modifying an element inside a container in place makes both the container and the
modified element output candidates.
- Operations like retrieving data, describing data, visualizing data, accessing
properties, and creating a copy are not considered as modifying the object in
place.

Step 2: extract outputs by filtering out output candidates which are not used in
any subsequent cells.

Step 3: add custom defined functions and classes as "defines_code", used functions
and classes as "refers_code" to each cell. A function or class is considered custom
defined if it is defined in the cell. A function or class is considered used if it
is called in the cell. Note that modules and builtin function names should not be
included.

Step 4: return a JSON string.

------
Given a Python Jupyter Notebook:
...[omitted due to space limitations]...
Cell-3:
```python
len(phone_data.query(ec)) / len(phone)
```

...[omitted due to space limitations]...

Think it step by step and return the JSON string as described above. Do not include
any code cell content in the answer.
```

Figure 14: An example baseline prompt (in part). The real prompt includes all code cells.

| NotebookID | C1 | C2 | C3 |
|---|---|---|---|
| jhoward/is-it-a-bird-creating-a-model-from-your-own-data | ✓ | ✓ | ✗ |
| kanncaa1/data-sciencetutorial-for-beginners | ✗ | | |
| ldfreeman3/a-data-science-framework-to-achieve-99-accuracy | ✓ | ✓ | ✗ |
| arthurtok/introduction-to-ensembling-stacking-in-python | ✓ | ✓ | ✗ |
| dansbecker/how-models-work | ✗ | | |
| jhoward/jupyter-notebook-101 | ✗ | | |
| sudalairajkumar/winning-solutions-of-kaggle-competitions | ✗ | | |
| jhoward/getting-started-with-nlp-for-absolute-beginners | ✗ | | |
| colinmorris/functions-and-getting-help | ✗ | | |
| thebrownviking20/everything-you-can-do-with-a-time-series | ✗ | | |
| kanncaa1/deep-learning-tutorial-for-beginners | ✗ | | |
| yogeshtak/exercise-your-first-machine-learning-model | ✗ | | |
| shivamb/data-science-glossary-on-kaggle | ✗ | | |
| kanncaa1/machine-learning-tutorial-for-beginners | ✗ | | |
| alexisbcook/getting-started-with-kaggle | ✗ | | |
| colinmorris/booleans-and-conditionals | ✗ | | |
| gzuidhof/full-preprocessing-tutorial | ✗ | | |
| therealcyberlord/coronavirus-covid-19-visualization-prediction | ✓ | ✗ | |
| ibtesama/getting-started-with-a-movie-recommendation-system | ✓ | ✗ | |
| ybifoundation/understanding-grammar-of-matplotlib | ✗ | | |
| residentmario/creating-reading-and-writing | ✗ | | |
| imdevskp/covid-19-analysis-visualization-comparisons | ✓ | ✗ | |
| tanulsingh077/twitter-sentiment-extaction-analysis-eda-and-model | ✓ | ✓ | ✗ |
| ybifoundation/create-series-pandas-dataframe | ✗ | | |
| ybifoundation/pandas-basic-functions | ✗ | | |
| kanncaa1/pytorch-tutorial-for-deep-learning-lovers | ✗ | | |
| dansbecker/submitting-from-a-kernel | ✗ | | |
| colinmorris/lists | ✗ | | |
| jhoward/how-does-a-neural-net-really-work | ✗ | | |
| ybifoundation/sklearn-inbuild-data-sets-for-machine-learning | ✗ | | |
| dansbecker/basic-data-exploration | ✗ | | |
| masumrumi/a-statistical-analysis-ml-workflow-of-titanic | ✓ | ✗ | |
| ybifoundation/seaborn-inbuild-data-sets-for-machine-learning | ✗ | | |
| anokas/data-analysis-xgboost-starter-0-35460-lb | ✓ | ✗ | |
| shahules/basic-eda-cleaning-and-glove | ✓ | ✗ | |
| colinmorris/loops-and-list-comprehensions | ✗ | | |
| rohanrao/tutorial-on-reading-large-datasets | ✗ | | |
| ybifoundation/download-dataset-from-openml-org | ✗ | | |
| ybifoundation/train-test-split | ✗ | | |
| vbmokin/data-science-for-tabular-data-advanced-techniques | ✗ | | |
| ybifoundation/generate-synthetic-data-using-sklearn | ✗ | | |
| ybifoundation/introduction-to-google-colab-notebook | ✗ | | |
| ybifoundation/pandas-install-update-and-help | ✗ | | |
| jhoward/linear-model-and-neural-net-from-scratch | ✓ | ✗ | |
| ybifoundation/encoding | ✗ | | |
| prashant111/a-guide-on-xgboost-hyperparameters-tuning | ✓ | ✗ | |
| colinmorris/working-with-external-libraries | ✗ | | |
| dansbecker/explore-your-data | ✗ | | |
| subinium/simple-matplotlib-visualization-tips | ✗ | | |
| colinmorris/strings-and-dictionaries | ✗ | | |
| fabiendaniel/customer-segmentation | ✓ | ✗ | |
| maksimeren/covid-19-literature-clustering | ✓ | ✓ | ✗ |
| kanncaa1/seaborn-tutorial-for-beginners | ✗ | | |
| artgor/eda-and-models | ✓ | ✗ | |

Table 5: Reasons for discarding a top voted notebook. These notebooks can be accessed via https://www.kaggle.com/code/{NotebookID} on Feb. 25, 2025. C1, C2, C3 are criteria 1, 2, and 3. "✓" is used when a criterion is met; "✗" is used when a criterion is violated.

| ID | NotebookID | C | L | F | D |
|---|---|---|---|---|---|
| 01 | alexisbcook/titanic-tutorial | 6 | 43 | 4 | 4 |
| 02 | pmarcelino/comprehensive-data-exploration-with-python | 32 | 127 | 32 | 108 |
| 03 | startupsci/titanic-data-science-solutions | 52 | 256 | 133 | 576 |
| 04 | serigne/stacked-regressions-top-4-on-leaderboard | 65 | 332 | 91 | 773 |
| 05 | yassineghouzam/introduction-to-cnn-keras-0-997-top-6 | 24 | 213 | 35 | 118 |
| 06 | ldfreeman3/a-data-science-framework-to-achieve-99-accuracy | 35 | 1111 | 103 | 146 |
| 07 | janiobachmann/credit-fraud-dealing-with-imbalanced-datasets | 63 | 928 | 153 | 442 |
| 08 | willkoehrsen/start-here-a-gentle-introduction | 62 | 666 | 94 | 424 |
| 09 | tanulsingh077/deep-learning-for-nlp-zero-to-transformers-bert | 43 | 285 | 105 | 317 |
| 10 | ash316/eda-to-prediction-dietanic | 71 | 358 | 117 | 461 |
| 11 | yassineghouzam/titanic-top-4-with-ensemble-modeling | 76 | 439 | 113 | 922 |
| 12 | dansbecker/your-first-machine-learning-model | 9 | 29 | 10 | 25 |
| 13 | ybifoundation/stars-classification | 20 | 38 | 22 | 57 |
| 14 | dansbecker/model-validation | 3 | 38 | 5 | 2 |
| 15 | abhishek/approaching-almost-any-nlp-problem-on-kaggle | 46 | 566 | 137 | 187 |
| 16 | dansbecker/underfitting-and-overfitting | 3 | 31 | 5 | 2 |
| 17 | gunesevitan/titanic-advanced-feature-engineering-tutorial | 46 | 644 | 84 | 494 |
| 18 | karnikakapoor/customer-segmentation-clustering | 29 | 212 | 40 | 205 |
| 19 | faressayah/stock-market-analysis-prediction-using-lstm | 27 | 282 | 59 | 54 |
| 20 | benhamner/python-data-visualizations | 15 | 78 | 14 | 14 |
| 21 | kanncaa1/feature-selection-and-data-visualization | 29 | 205 | 47 | 79 |
| 22 | jagangupta/time-series-basics-exploring-traditional-ts | 40 | 369 | 35 | 85 |
| 23 | gusthema/house-prices-prediction-using-tfdf | 24 | 100 | 23 | 81 |
| 24 | apapiu/regularized-linear-models | 29 | 80 | 41 | 131 |
| 25 | dansbecker/random-forests | 2 | 27 | 4 | 1 |
| 26 | dansbecker/submitting-from-a-kernel | 3 | 27 | 4 | 3 |
| 27 | nadintamer/titanic-survival-predictions-beginner | 41 | 286 | 96 | 369 |
| 28 | ybifoundation/simple-linear-regression | 18 | 28 | 23 | 61 |
| 29 | kanncaa1/plotly-tutorial-for-beginners | 21 | 439 | 18 | 22 |
| 30 | rtatman/data-cleaning-challenge-handling-missing-values | 16 | 49 | 12 | 15 |
| 31 | residentmario/indexing-selecting-assigning | 22 | 26 | 21 | 22 |
| 32 | ybifoundation/fish-weight-prediction | 20 | 31 | 24 | 57 |
| 33 | ybifoundation/cancer-prediction | 21 | 31 | 26 | 64 |
| 34 | ybifoundation/chance-of-admission | 21 | 31 | 26 | 64 |
| 35 | ybifoundation/multiple-linear-regression | 21 | 34 | 26 | 64 |
| 36 | omarelgabry/a-journey-through-titanic | 20 | 289 | 31 | 93 |
| 37 | ybifoundation/purchase-prediction-micronumerosity | 21 | 35 | 26 | 65 |
| 38 | dansbecker/handling-missing-values | 5 | 58 | 17 | 7 |
| 39 | ybifoundation/credit-card-default | 22 | 33 | 27 | 65 |
| 40 | ybifoundation/tutorial-decision-tree-regression-scikit-learn | 29 | 73 | 42 | 148 |
| 41 | deffro/eda-is-fun | 40 | 168 | 46 | 70 |
| 42 | gpreda/santander-eda-and-prediction | 61 | 329 | 111 | 160 |
| 43 | kashnitsky/topic-1-exploratory-data-analysis-with-pandas | 39 | 79 | 37 | 125 |
| 44 | kushal1996/customer-segmentation-k-means-analysis | 27 | 188 | 37 | 42 |
| 45 | kenjee/titanic-project-example | 58 | 390 | 115 | 381 |
| 46 | christofhenkel/how-to-preprocessing-when-using-embeddings | 25 | 123 | 30 | 82 |
| 47 | sinakhorami/titanic-best-working-classifier | 13 | 157 | 31 | 56 |
| 48 | shrutimechlearn/step-by-step-diabetes-classification | 36 | 153 | 53 | 127 |
| 49 | ybifoundation/binary-classification | 21 | 31 | 26 | 64 |
| 50 | ybifoundation/multi-category-classification | 21 | 31 | 26 | 64 |

Table 6: Individual notebook statistics. These notebooks can be accessed via https://www.kaggle.com/code/{NotebookID} on Feb. 25, 2025. The C column denotes number of code cells, the L column denotes number of lines, the F column denotes number of information flows, and the D column denotes number of transitive dependencies.

```
Given a Python program block, determine if an object is an input. An input is a
variable that is used in the program block but not defined within it, or variables
used before being reassigned.

Important Cases:
1. Conditional Statement Within a Loop
A variable may be initialized during an early iteration of a loop and then utilized
in subsequent iterations. In such cases, the variable is not considered an input,
as its value originates from the loop's execution rather than external sources.
2. Shared References (Aliased Variables)
- If multiple variables reference the same object (e.g., through assignment or
being stored inside a data structure), and one of them is an input, then all
variables referring to that object are also inputs.
- If a container is an input, then all elements inside the container are also
inputs.
- If an element inside a container is an input, then the container itself is also
an input.

Preceding context (for shared reference):
```python
datasets = [train, test]
```

Question:
In the following Python program block, is "train" an input? Return the final answer
in the form "Yes(No), "{x}" is (not) an input" where "{x}" is the actual name of
the variable.
```python
for dataset in datasets:
    dataset.dropna(inplace=True)
```

Answer:
```

Figure 15: An example CRABS prompt for resolving ambiguous inputs

| Method | Information Flows | | | | |
|---|---|---|---|---|---|
| | Prec.(%) | Rec.(%) | $F_1$(%) | Acc.(%) | EM(%) |
| CRABS | **98.59** | **98.52** | **98.56** | **97.30** | **73** |
| w/o syntactic phase | 95.28 | 89.49 | 91.18 | 86.49 | 40 |
| w/o cell-by-cell prompting | 92.19 | 89.92 | 90.79 | 85.39 | 40 |

Table 7: Information flows: ablation study results for 45 selected notebooks.

| Method | Transitive Dependencies | | | | |
|---|---|---|---|---|---|
| | Prec.(%) | Rec.(%) | $F_1$(%) | Acc.(%) | EM(%) |
| CRABS | **99.42** | **99.95** | **99.67** | **99.37** | **82** |
| w/o syntactic phase | 95.80 | 87.88 | 89.20 | 84.13 | 42 |
| w/o cell-by-cell prompting | 89.96 | 77.40 | 82.22 | 77.37 | 36 |

Table 8: Transitive dependencies: ablation study results for 50 notebooks.

| Method | Transitive Dependencies | | | | |
|---|---|---|---|---|---|
| | Prec.(%) | Rec.(%) | $F_1$(%) | Acc.(%) | EM(%) |
| CRABS | 99.36 | **99.94** | **99.64** | **99.30** | **80** |
| w/o syntactic phase | 95.60 | 88.50 | 89.31 | 84.49 | 44 |
| w/o cell-by-cell prompting | **99.96** | 86.00 | 91.36 | 85.97 | 40 |

Table 9: Transitive dependencies: ablation study results for 45 selected notebooks.

```
Given a Python program block, determine if an object is an output candidate. An
output candidate is a variable that is defined, updated, or mutated in the program
block.

Important Cases:
1. Method Calls (object.method()) or Function Calls (function(object))
- If a method (or a function) modifies the object in place (e.g., updating data,
changing the structure, or adjusting properties in place), the object is an output
candidate.
- If a method (or a function) does not modify the object (e.g., retrieving data,
describing data, visualizing data, or accessing properties), the object is not an
output candidate.
- If a method (or a function) returns a new object (e.g., creating a copy) without
modifying the original, the original object is not an output candidate.
2. Reassignment (variable = ...)
If a variable is assigned a new value, it is an output candidate.
3. Shared References (Aliased Variables)
- If multiple variables reference the same mutable object (e.g., through assignment
or being stored inside a data structure), modifying the object in place through one
reference makes all references to that object output candidates.
- Modifying the container in place makes only the container an output candidate,
but not its elements.
- Modifying an element inside a container in place makes both the container and the
modified element output candidates.
- Operations like retrieving data, describing data, visualizing data, accessing
properties, and creating a copy are not considered as modifying the object in
place.

Preceding context (for shared reference):
```python
datasets = [train, test]
```

Question:
In the following Python program block, is "datasets" an output candidate? Return
the final answer in the form "Yes(No), "{x}" is (not) an output candidate" where "
{x}" is the actual name of the variable.
```python
for dataset in datasets:
    dataset.dropna(inplace=True)
```

Answer:
```

Figure 16: An example CRABS prompt for resolving ambiguous output candidates

```
Given a Python program block, return a JSON string with "inputs",
"output_candidates", "defines_code", "refers_code", and "shared_references" keys.
Do not include any names of built-in functions and variables.
- "inputs" are variables that are used in the program block but not defined within
it, or variables used before being reassigned. Besides shared references, one
important case is about conditional statement within a loop, i.e., when a variable
is initialized during an early iteration of a loop and then utilized in subsequent
iterations. In such cases, the variable is not considered an input, as its value
originates from the loop's execution rather than external sources.
- "output_candidates" are variables that are defined, updated, or mutated in the
program block. Besides shared references, there are two important cases:
     1. Method Calls (object.method()) or Function Calls (function(object))
         - If a method (or a function) modifies the object in place (e.g., updating
data, changing the structure, or adjusting properties in place), the object is an
output candidate.
         - If a method (or a function) does not modify the object (e.g., retrieving
data, describing data, visualizing data, or accessing properties), the object is
not an output candidate.
         - If a method (or a function) returns a new object (e.g., creating a copy)
without modifying the original, the original object is not an output candidate.
     2. Reassignment (variable = ...)
         - If a variable is assigned a new value, it is an output candidate.
- "defines_code" is a list of functions or classes defined in the program block.
- "refers_code" is a list of user-defined functions or classes used in the program
block, selected from a given list.
- "shared_references" is a string, which contains descriptions about the shared
references (aliased variables) based on previous context and current program block.
If there are no shared references, return "NA". Otherwise, it must explicitly
include:
     - How multiple variables reference the same object (e.g., through assignment or
being stored inside a data structure).
     - The relationship in both directions, meaning if one variable is modified, how
it affects the others and vice versa.

Previous context:
```python
The variables 'train' and 'test' are stored inside the list 'datasets'. If 'train'
or 'test' is modified, the change will be reflected in 'datasets' as well, and vice
versa, since 'datasets' holds references to these objects.
```

Current program block:
```python
for dataset in datasets:
    dataset.dropna(inplace=True)
```

From the following given list, select only the functions and classes that are used
and store them in "refers_code".
Given functions and classes:
[]

JSON string: (Think it step by step, then return the final results only)
```

Figure 17: An example prompt for the ablation study S1 (i.e., w/o syntactic phase).

```
Given a Python Jupyter Notebook with definite and possible annotations, extract the
input and output variables for each code cell and return a JSON string.

Each dictionary in the JSON list represents a code cell and contains:
- "inputs": List of input variables.
- "outputs": List of output variables.

Rules for Extraction:
Step 1: Include definite inputs and outputs
1. Add all definite inputs to the final inputs list.
2. Add all definite outputs to the final outputs list.

Step 2: Process ambiguous inputs
1. Compute ambiguous inputs by excluding definite inputs from possible inputs.
2. Add an ambiguous input only if it follows these rules:
- It is used in the code cell but not defined within it.
- It is used before reassignment in the same cell.
Special Cases:
- Conditional Statement Within a Loop: If a variable is initialized during an early
iteration of a loop and then utilized in subsequent iterations, it is not an input.
- Aliased/shared references:
    - If multiple variables reference the same object (directly or via a
container), all variables referring to that object are also inputs.
    - If an container is an input, then all elements inside the container are also
inputs.
    - If an element inside a container is an input, then the container itself is
also an input.

Step 3: Process ambiguous outputs
1. Compute ambiguous outputs by excluding definite outputs from possible outputs.
2. Add an ambiguous output only if it follows these rules:
- It is defined, updated, or mutated within the code cell.
Special Cases:
- Method Calls & Functions: If a function or method modifies an object in place,
the object is an output.
- Reassignment: If a variable is reassigned, it is an output.
- Aliased/shared references:
    - If a mutable object is modified in place, all references to it should be
outputs.
    - Modifying the container in place makes only the container an output
candidate, but not its elements.
    - Modifying an element inside a container in place makes both the container and
the modified element output candidates.

Step 4: Filter outputs
Remove output candidates that are not used as inputs in any subsequent code cell.

Step 5: Return the JSON Output
- Format the final list as a JSON string.
- Maintain the same order of code cells as in the notebook.

------
Given a Python Jupyter Notebook with annotations:
...[omitted due to space limitations]...
Cell-3:
``` python
# definite inputs: [ec, phone]; outputs: []
# possible inputs: [ec, phone]; outputs: [ec, phone]

len(phone.query(ec)) / len(phone)
```

...[omitted due to space limitations]...

Think it step by step and return the JSON string as described above. Do not include
any code cell content in the answer.
```

Figure 18: Part of an example prompt for the ablation study S2 (i.e., w/o cell-by-cell prompting). The real prompt includes all code cells.

| ID | Information Flows (%) | | | | Transitive Dependencies (%) | | | |
|----|----------|-------|-------|-------|----------|-------|-------|-------|
| | baseline | CRABS | S1 | S2 | baseline | CRABS | S1 | S2 |
| 01 | 100 | 100 | 100 | 100 | 100 | 100 | 100 | 100 |
| 02 | 0 | 100 | 100 | 100 | 0 | 100 | 100 | 100 |
| 03 | 64.25 | 96.58 | 57.29 | 60.34 | 67.59 | 99.91 | 32.80 | 58.13 |
| 04 | 0 | 100 | 97.24 | 94.51 | 0 | 100 | 78.84 | 95.55 |
| 05 | 91.43 | 97.14 | 85.71 | 91.43 | 83.17 | 100 | 85.33 | 83.17 |
| 06 | 59.74 | 94.17 | 65.03 | 61.11 | 92.65 | 98.65 | 58.25 | 81.30 |
| 07 | 90.34 | 93.46 | 98.00 | 94.77 | 91.67 | 97.08 | 94.64 | 92.06 |
| 08 | 0 | 91.49 | 92.82 | 96.81 | 0 | 99.30 | 96.21 | 99.88 |
| 09 | 0 | 94.29 | 94.58 | 89.52 | 0 | 98.14 | 89.74 | 83.03 |
| 10 | 0 | 100 | 95.54 | 0 | 0 | 100 | 72.65 | 0 |
| 11 | 0 | 100 | 100 | 0 | 0 | 100 | 100 | 0 |
| 12 | 100 | 90 | 100 | 100 | 100 | 100 | 100 | 100 |
| 13 | 100 | 100 | 97.67 | 95.45 | 100 | 100 | 99.12 | 96.36 |
| 14 | 100 | 100 | 100 | 100 | 100 | 100 | 100 | 100 |
| 15 | 0 | 98.54 | 97.42 | 100 | 0 | 99.47 | 95.41 | 100 |
| 16 | 100 | 100 | 100 | 100 | 100 | 100 | 100 | 100 |
| 17 | 43.24 | 93.41 | 83.44 | 0 | 45.78 | 100 | 98.87 | 0 |
| 18 | 0 | 90 | 87.32 | 100 | 0 | 99.03 | 82.18 | 100 |
| 19 | 73.79 | 100 | 68.09 | 70.21 | 97.14 | 100 | 88.66 | 95.15 |
| 20 | 100 | 100 | 100 | 100 | 100 | 100 | 100 | 100 |
| 21 | 98.92 | 93.62 | 96.70 | 97.87 | 100 | 92.40 | 100 | 98.06 |
| 22 | 87.10 | 100 | 95.65 | 0 | 90.32 | 100 | 94.74 | 0 |
| 23 | 73.91 | 100 | 100 | 73.91 | 66.12 | 100 | 100 | 66.12 |
| 24 | 89.16 | 100 | 95.12 | 100 | 91.23 | 100 | 89.29 | 100 |
| 25 | 100 | 100 | 100 | 100 | 100 | 100 | 100 | 100 |
| 26 | 66.67 | 100 | 85.71 | 100 | 100 | 100 | 100 | 100 |
| 27 | 94.62 | 99.48 | 91.01 | 92.55 | 99.73 | 100 | 40.95 | 90.50 |
| 28 | 86.96 | 100 | 100 | 86.96 | 85.98 | 100 | 100 | 85.98 |
| 29 | 100 | 100 | 100 | 100 | 100 | 100 | 100 | 100 |
| 30 | 100 | 100 | 59.26 | 100 | 100 | 100 | 75.00 | 100 |
| 31 | 100 | 100 | 66.67 | 100 | 100 | 100 | 84.62 | 100 |
| 32 | 87.50 | 100 | 100 | 87.50 | 86.00 | 100 | 100 | 86.00 |
| 33 | 88.46 | 100 | 98.04 | 88.46 | 86.73 | 100 | 99.21 | 86.73 |
| 34 | 88.46 | 100 | 100 | 88.46 | 86.73 | 100 | 100 | 86.73 |
| 35 | 88.46 | 100 | 100 | 88.46 | 86.73 | 100 | 100 | 86.73 |
| 36 | 100 | 100 | 82.86 | 93.55 | 100 | 100 | 85.19 | 95.51 |
| 37 | 100 | 100 | 100 | 88.46 | 100 | 100 | 100 | 87.93 |
| 38 | 90.32 | 100 | 100 | 100 | 100 | 100 | 100 | 100 |
| 39 | 88.89 | 100 | 100 | 88.89 | 86.96 | 100 | 100 | 86.96 |
| 40 | 78.57 | 100 | 97.56 | 78.57 | 79.67 | 100 | 76.15 | 79.67 |
| 41 | 0 | 100 | 100 | 100 | 0 | 100 | 100 | 100 |
| 42 | 70.87 | 100 | 83.78 | 90.09 | 88.50 | 100 | 82.35 | 97.11 |
| 43 | 98.63 | 100 | 83.33 | 0 | 100 | 100 | 74.40 | 0 |
| 44 | 100 | 100 | 98.63 | 100 | 100 | 100 | 98.80 | 100 |
| 45 | 0 | 100 | 93.45 | 100 | 0 | 100 | 87.40 | 100 |
| 46 | 69.39 | 100 | 96.77 | 93.33 | 59.83 | 100 | 94.25 | 87.67 |
| 47 | 60.38 | 100 | 27.78 | 26.67 | 100 | 100 | 13.33 | 40 |
| 48 | 98.11 | 96.23 | 92.45 | 90.57 | 99.22 | 99.61 | 93.04 | 91.45 |
| 49 | 88.46 | 100 | 100 | 88.46 | 86.73 | 100 | 100 | 86.73 |
| 50 | 88.46 | 100 | 96.00 | 88.46 | 86.73 | 100 | 98.41 | 86.73 |

Table 10: Detailed results for notebook understanding task. The S1 and S2 columns denote ablation study S1 (i.e., w/o syntactic phase) and ablation study S2 (i.e., w/o cell-by-cell prompting). Values indicate the $F_1$ score of a notebook, where the notebook can be referred to via the ID column and Table 6.

