# OpenReview forum: "CRABS: A syntactic-semantic pincer strategy for bounding LLM interpretation of Python notebooks"
_colmweb.org/COLM/2025/Conference — COLM 2025_

### Official Review · Reviewer_TYCf · 2025-05-04

**Rating:** 7
**Confidence:** 4
**Ethics Flag:** 1

**Summary:**

This research paper presents a study on understanding Python notebook execution dependencies using the CRABS (Cell-by-cell Reasoning About Python notebook Structure) strategy. The empirical foundation of the work is generally solid, built upon a dataset of 50 annotated Python notebooks from Kaggle that were specifically selected for their relevance to machine learning workflows. The authors employed a well-defined experimental procedure using a cell-by-cell prompting strategy that effectively addresses long-context challenges and hallucinations in large language models. However, the study's methodological rigor is somewhat undermined by the lack of a detailed explanation regarding the sampling strategy for selecting the most up-voted notebooks, which may introduce selection bias and limit the generalizability of findings.
The technological contribution of this work is significant, with the CRABS strategy demonstrating clear effectiveness in addressing the complex problem of understanding notebook execution dependencies without actual code execution. The solution shows substantial performance improvements across multiple metrics, including a 28 percentage-point increase in F1 score, 32 points in accuracy, and 48 points in the exact match for information flow graphs compared to baseline approaches. The innovative combination of neural and symbolic methods represents a promising approach to resolving ambiguities in code analysis, though concerns remain about the scalability and broader applicability of the approach beyond the specific domain tested.
From a forward-looking perspective, the research addresses a crucial gap in understanding Python notebooks without execution, which is essential for improving usability and adaptability in data science workflows. The study design demonstrates innovation by leveraging pre-trained language models to resolve data flow ambiguities between notebook cells, contributing to the broader field of automated code analysis.
While the technical content is robust, the manuscript suffers from presentation issues that detract from its overall impact. The paper contains frequent grammatical errors, particularly related to verb tense consistency and subject-verb agreement, indicating a need for thorough proofreading. These clarity issues, while minor individually, collectively impact the professional presentation of otherwise solid research contributions.

**Questions To Authors:**

- The methodology does not address potential biases introduced by the zero-shot in-context learning approach, which may affect the reliability of the LLM's ambiguity resolution. Suggestion: Provide a clear rationale and criteria for notebook selection to ensure representativeness and reduce bias
- The experimental procedure does not include control conditions to compare the effectiveness of the cell-by-cell prompting strategy against other potential strategies, limiting the robustness of the findings. Suggestion: Implement control experiments using different prompting strategies to establish a comparative baseline
- The validity of the syntactic parsers used to bound the ground truth is not thoroughly evaluated, which may impact the accuracy of the notebook understanding task. Suggestion: Conduct a validation study to assess the accuracy and reliability of the syntactic parsers used in bounding the ground truth
- While the manuscript focuses on resolving ambiguities in Python notebooks, it does not explore the broader implications of LLMs in transforming data science roles and education
- The manuscript does not extensively address the potential for LLMs to handle context-dependent translations. Suggestion: Integrate context-aware parsing techniques to enhance the CRABS strategy's ability to manage context-dependent data flows in Python notebooks
- The manuscript's approach relies heavily on syntactic parsing, which may not fully capture semantic nuances or context-dependent information flows, a limitation that may be addressed by dynamic prompting

**Reasons To Accept:**

This paper merits acceptance due to its innovative approach and significant technological contributions. The research presents the CRABS strategy, which effectively addresses the complex challenge of understanding notebook execution dependencies without requiring actual execution—a crucial advancement for improving data science workflow usability. The methodology demonstrates technical excellence through its robust data collection process utilizing 50 annotated Python notebooks from Kaggle specifically chosen for machine learning relevance. The cell-by-cell prompting strategy successfully mitigates long-context challenges and hallucinations, resulting in impressive performance improvements: a 28 percentage-point increase in F1 score, 32 in accuracy, and 48 in exact match for information flow graphs compared to baseline approaches. Particularly noteworthy is the paper's principled approach combining neural and symbolic methods, representing a promising general methodology for resolving ambiguities in code analysis. While there are minor grammatical issues requiring proofreading and some limitations regarding sampling strategy explanation and broader generalizability beyond Python notebooks, these do not diminish the paper's substantial contributions to the field. The research fills a specific gap in computational understanding of non-executed notebooks, making it a valuable addition to the literature that deserves publication.

**Reasons To Reject:**

Despite the paper's promising approach to notebook understanding, we should consider several critical methodological limitations. The study fails to address potential biases introduced by the zero-shot in-context learning approach, which significantly undermines the reliability of the LLM's ambiguity resolution capability. Furthermore, the experimental design lacks essential control conditions to compare the cell-by-cell prompting strategy against alternative approaches, severely limiting the robustness of the findings. Another major concern is the unvalidated nature of the syntactic parsers used to bound the ground truth, potentially compromising the accuracy of the entire notebook understanding task. The heavy reliance on syntactic parsing without adequately capturing semantic nuances represents a fundamental weakness, as does the absence of context-aware parsing techniques to manage context-dependent data flows. Finally, the manuscript misses valuable opportunities to explore the broader implications of LLMs in transforming data science roles and education. These substantial shortcomings collectively indicate that the research requires significant refinement before it can make a meaningful contribution to the field.

---

> ### Author Response · Authors · 2025-06-03
>
> Thank you for your valuable comments and suggestions! We address your questions below. Please let us know if you have any follow-up questions.
>
> **Answer to Q1**: Please refer to the "clarification of notebook selection" section in the general responses.
>
> **Answer to Q2**: Our comparisons are limited to two ablation studies: (1) cell-by-cell prompting without providing any hints from syntactic parsers; and (2) a single prompt for the whole notebook with hints for all cells from the syntactic parsers. Table 3 shows the percentage-point decreases of 7 for (1) and 16 for (2) in F1 score. We agree with the value of looking into other prompting strategies. For example, in future work we plan to try an approach of including in the prompts a sliding window of preceding and succeeding code cells.
>
> **Answer to Q3**: Please refer to the "clarification of the role of syntactic parsers" section in the general responses. Our syntactic parsers employ the Python built-in ast module to access and traverse the abstract syntax tree for each cell, a widely employed technique. Via the traversals of the AST we produce lower and upper estimates of inter-cell I/O sets that bound the ground truth represented by our notebooks. We believe this covers most of the cases in practice. However, we acknowledge that the validity of our syntactic parsers are not thoroughly evaluated, and that a practical implementation of CRABS will require comprehensive validation and testing.
>
> **Answer to Q4**:  We agree that LLMs have the potential to significantly transform data science roles and education. Our current focus is on the notebook understanding task and the CRABS strategy for performing it, but we plan to explore the broader implications of CRABS and related approaches for enhancing the transparency of notebooks and the scientific research that depend on them.
>
> **Answer to Q5 & Q6**: Thank you for these suggestions. We will certainly pursue these opportunities in our future work.

---

> > ### Comment · Reviewer_TYCf · 2025-06-04
> >
> > Thank the authors for their response.
> > I hope the authors plan to expand their work in the future.

---

### Official Review · Reviewer_TcJh · 2025-05-10

**Rating:** 7
**Confidence:** 3
**Ethics Flag:** 1

**Summary:**

This paper describes CRABS (Capture and Resolve Assisted Bounding Strategy), an algorithm that parses the information flow and execution dependencies between code cells in Python Notebooks based on AST parser and zero-shot prompting of LLMs. The authors manually annotated the 50 high-quality and highly-up-voted public notebooks and use the dataset as the basis for evaluation of the CRABS algorithm and comparison with baselines. CRABS is a hybrid approach that combines deterministic, explainable ASR parsing with LLM's pattern understanding capabilities. It shows significantly improved parsing accuracy compared to the baselines, as demonstrated by using the precision/recall/F1 metrics for data dependency edges and the overall exact-match (EM) rate.

The paper is well written and easy to read and follow. It clearly explains the background and importance of this problem (dependency parsing in Python notebooks) and why it is a nontrivial problem, i.e., challenges such as unclear side effect of Python function calls and hidden modifications of values within data structures such as Python lists. These challenges render the problem beyond the reach of traditional, non-ML algorithms, at least without the execution of the code. The authors also explained the shortcomings of previous effort at tackling this problem using LLMs, including the tendency to hallucinate nonexistent information flow and dependencies.

The originality of the authors' approach (CRABS) is apparent but not to the degree of groundbreaking. The CRABS strategy can be briefly summarized as "divide and conquer", i.e., augmenting AST analysis with prompted LLMs at the level of individual code cells and individual variables and methods, instead of relying solely on an LLM to analyze the notebook as a whole. This hybrid approach achieves clear advantage over the LLM-only baseline as well as over the AST-only baseline, which is a significant contribution.

**Questions To Authors:**

Q1. See my weakness point W2 above: are the annotation of the dependency graph of a notebook unique, given the possibility of multiple parallel upstream dependencies of a given code cell? If not, how was that possibility accommodated? Were all the options in the parallel set of dependencies considered dependencies?

Q2: Also related to W2 above: How did the authors ensure the quality of the manual annotation? Was there any measurement of inter-rater reliability?

**Reasons To Accept:**

Strengths of this paper that argue for acceptance.

S1. This paper focuses on an important problem in AI-assisted code analysis that has important practical applications in data science and other applicable domains of Python Notebooks.

S2. The authors constructed a dataset with manually-annotated ground truth to enable evaluation of the CRABS algorithm and the contrasting baselines. The authors selected a set of clear and informative evaluation metrics, including precision/recall/F1 values for information flow between code cells and the EM rate on the overall notebook.

S3. A moderately innovative algorithm that combines the strengths of syntactic parsing (AST) and LLM (gpt-4o as used by the authors), which achieved the highest accuracy among all tested solutions. An overall EM rate of 74% was achieved on the information flow graph; 84% for the transitive dependency. The F1 scores on the cell-wise information flow were shown to be at 97% or higher.

S4. The authors performed ablation studies to demonstrate and quantify the contribution of different components of the CRABS algorithm: AST parser and cell-by-cell prompting.

**Reasons To Reject:**

Weaknesses of the paper that argue against acceptance.

W1. The dataset that the author constructed is relatively small in size (50 notebooks) and skewed towards the high-quality end of the space of all possible Python notebooks. For practical applications, many users (such as typical data scientists and ML engineers) will perform dependency analyses on their own notebooks, which may have lower code quality and messier (more complex) dependencies compared to the high-quality ones. This casts some doubt on the generalizability of the results to "average quality" notebooks.

W2. The authors did not characterize the manual annotation process in several aspects: a) whether the annotation result is unique. For instance, whether multiple code cells can each be considered the execution dependency of a downstream code cell. In those cases, how did the annotation result accommodate this possibility? b) The reliability of the manual annotation process, e.g., whether some of the notebooks were annotated by multiple human annotators and the agreements among them quantified. c) Are the annotation results be made available so that future studies can build on the dataset and compare their results with the current paper?

W3. The authors relied solely on zero-shot prompting in CRABS. It is possible that using few-shot examples can further improve the accuracy of CRABS, but the authors did not mention or discuss that in the paper. The same goes for the potential approach of LoRA tuning the foundational model on a reasonable amount of annotated data.

W4. This paper does not report the computational cost of CRABS and how it compares with the baseline methods including AST (which is expected to be very small in comparison) and the whole-notebook prompting. This question is especially relevant given the large number of code cells (30 on average) and information flow (49 on average) within each notebook according to Table 1. For practical application of such a system, the users of the Python notebooks would be editing and running the notebook in real time. Therefore the latency matters.

---

> ### Author Response · Authors · 2025-06-03
>
> Thank you for your valuable feedback! We address your comments below. Please let us know if you have any follow-up questions.
>
> **Answer to W1**: Please refer to the "clarification of notebook selection" section in the general responses.
>
> **Answer to W2 & Q1 & Q2**: Yes, the annotation result is unique, and there is only one correct dependency graph per notebook.
>
> a) Multiple code cells can each be considered the execution dependency of a downstream code cell. Taking Figure 3 as an example, cell 3 uses the phone data from cell 1 and the ec string from cell 2. There is only one unique information flow graph (Figure 3) and one unique cell execution dependency graph (Figure 4) for the notebook (Figure 2). We annotate this example notebook by inputs and outputs per code cell (see below), which can be used to construct the corresponding graphs.
> ```
> [
>   {"id": 1, "outputs": ["phone", "survey"]},
>   {"id": 2, "outputs": ["ec"]},
>   {"id": 3, "inputs": ["phone", "ec"]},
>   ...
> ]
> ```
> b) Annotation accuracy is ensured through careful validation. Ground truth is annotated by a skilled data scientist, who is confident in the majority of cell I/O sets. For uncertain cases, she compared the program state before and after cell execution to determine whether the data should be considered an output candidate.
>
> c) Yes, annotations will be made available together with code via GitHub.
>
> **Answer to W3**: Given the high performance of GPT-4o on this task (correctly resolving 98% of the ambiguities), we don’t think there is a lot of room for improvement. However, we do see the potential for using these techniques (e.g., few-shot learning and LoRA) to help improve the performance of smaller and cheaper LLMs (e.g., Qwen3 8B).
>
> **Answer to W4**: Thank you for highlighting the issue of computational cost. Pure AST approaches do not represent baselines but instead provide lower and upper estimates that bound the ground truth. Their metrics shown in Table 2 are not intended for evaluation, but for reflecting the characteristics of our notebooks. I will clarify this in the next version of the paper and use *lower estimate* and *upper estimate* to replace *MinIOParser* and *MaxIOParser*.
>
> Table R2 shows the computation cost per notebook (in seconds) using the baseline and CRABS approaches, and will be added to the appendix of the paper. The baseline approach is a single prompt for the whole notebook. Compared to baseline, we observe that (1) for notebooks with no (e.g., ID#1) or very few (e.g., ID#31) ambiguities left by syntactic parsers, CRABS requires less time and achieves comparable performance; (2) for complicated and lengthy notebooks with many ambiguities (e.g., ID#10), CRABS takes more time than baseline but yields better results; (3) for medium-complexity notebooks with around 10 ambiguities (e.g., ID#49, #50), CRABS requires similar time to baseline but also achieves better results.
>
>
>
> |ID|Baseline|CRABS|
> |---|----------|---------|
> | 1 | 2.585 | 0.002 |
> | 2 | 16.447 | 21.726 |
> | 3 | 31.378 | 107.717 |
> | 4 | 29.086 | 26.999 |
> | 5 | 12.941 | 8.776 |
> | 6 | 18.811 | 131.171 |
> | 7 | 39.09 | 82.632 |
> | 8 | 38.363 | 28.601 |
> | 9 | 30.928 | 46.433 |
> | 10 | 36.798 | 90.377 |
> | 11 | 32.821 | 60.888 |
> | 12 | 14.465 | 3.234 |
> | 13 | 13.569 | 10.278 |
> | 14 | 8.899 | 2.022 |
> | 15 | 31.229 | 82.1 |
> | 16 | 10.441 | 0.002 |
> | 17 | 32.274 | 44.712 |
> | 18 | 13.723 | 16.852 |
> | 19 | 18.417 | 46.558 |
> | 20 | 5.414 | 14.014 |
> | 21 | 18.234 | 39.767 |
> | 22 | 22.29 | 14.2 |
> | 23 | 12.073 | 12.079 |
> | 24 | 18.476 | 17.286 |
> | 25 | 7.43 | 0.001 |
> | 26 | 11.101 | 0.002 |
> | 27 | 17.3 | 58.398 |
> | 28 | 7.302 | 4.214 |
> | 29 | 11.88 | 4.297 |
> | 30 | 16.912 | 9.14 |
> | 31 | 13.146 | 2.714 |
> | 32 | 9.118 | 5.292 |
> | 33 | 10.223 | 7.522 |
> | 34 | 9.073 | 11.214 |
> | 35 | 8.762 | 8.321 |
> | 36 | 14.611 | 9.429 |
> | 37 | 10.88 | 10.101 |
> | 38 | 12.867 | 10.244 |
> | 39 | 11.453 | 8.798 |
> | 40 | 11.149 | 26.113 |
> | 41 | 12.691 | 37.584 |
> | 42 | 23.744 | 64.642 |
> | 43 | 11.169 | 29.093 |
> | 44 | 11.234 | 12.326 |
> | 45 | 28.294 | 62.894 |
> | 46 | 9.652 | 4.786 |
> | 47 | 8.355 | 24.846 |
> | 48 | 22.409 | 39.376 |
> | 49 | 6.896 | 6.802 |
> | 50 | 8.478 | 7.08 |
>
> Table R2: Computation cost per notebook (in seconds).

---

> > ### Comment · Reviewer_TcJh · 2025-06-07
> > **Follow-up comments**
> >
> > I thank the author for their reply to my comments. Please see my follow-up comments below.
> >
> > Re. the authors response to W2, it is good to clarify that in this study the dependency relation is treated as unique. However, I think this leads to a limitation that should be documented because in some notebooks the dependency relation might not be unique. Think of the following possibility: In a notebook, there are three code cells, A, B, and C. Cell A sets up an ML model of one kind (say an LSTM). Cell B sets up another ML model that has exactly the same input/output specs as the said model, but with a different model architecture (say a transformer). Cell C trains and evaluates the model, whether it is from cell A or B. So the user of the notebook can run the notebook either in the order of A->C or B->C, but it doesn't make much sense to run it in the order of A->B->C.
> >
> > This is an example of multiple, non-unique valid information flow paths in a notebook. It might not be common, but I think it certainly exists in reality. The paper should discuss this, perhaps as a limitation unless the author has a way to address this type of scenarios.
> >
> > Re. W4, please summarize the computation cost of the two approaches by reporting the summary statistics to make it easier to assess than the raw data.
> >
> > I will maintain my rating score for the time being.

---

> > > ### Author Response · Authors · 2025-06-09
> > >
> > > **Response to follow-up questions on W2**: Thank you for highlighting this aspect of understanding Jupyter notebooks in general. The example you provide illustrates very well both a limitation of CRABS and our immediate objective for the method. As you say (and as mentioned briefly in sections 3 and 4.1), we make the simplifying assumption, when extracting the information flow graph for the notebook, that the notebook is meant to be executed in its entirety from top to bottom; we do not consider (for now) partial executions of notebooks, out-of-order execution of cells, or manual stepping and skipping over cells (all of which indeed are common use cases of Jupyter notebooks).  In addition our aim, here, is to produce a single information flow graph for the notebook that includes all of the information flows that can occur during that top-to-bottom execution; where multiple paths of execution are possible–say due to conditional execution of cell contents based on data or parameter values–the information flow graph represents the union of the possible flows. This  is why we say this information flow graph extracted by CRABS is unique.  This approach follows that taken by YesWorkflow (McPhillips et al., 2015) where user-annotations reveal the information flows in a script.  For the example you give, the information flow graph we would aim to produce with CRABS would show the ML model flowing both from cell A to to cell C, and from cell B to cell C (this is how YesWorkflow would do it), with the understanding that in a particular execution only one of these flows might be realized. These are the information flows that can happen.  However, we indeed would like to develop methods for extracting and representing multiple, distinct information flow graphs such as those you describe through enhancements both to the parsing and LLM-prompting elements of CRABS and via visualizations that distinguish the flows that occur in every run of a notebook from those that occur in only some executions. We will add parts of the above discussion to Section 3 to clarify the aim and limitations of the current model, and to the new Future Work section to highlight the opportunities for enhancement along these lines.
> > >
> > >
> > > **Response to follow-up questions on W4**: Thank you very much for asking for more details about the computational cost of CRABS and the comparison with the baseline method. Further analysis of the performance data shows that the computational cost (quantified as the total of the latencies of the individual requests associated with analyzing a single notebook) is proportional to the total number of cell I/O ambiguities in the notebook; the least squares line has a regression constant of 0.94 sec/ambiguity and R-squared of 0.96. Least-squares fit of the baseline cost against the number of ambiguities is much poorer with an R-squared=0.54, but the regression constant of 0.22 sec/ambiguity is still informative: the point of intersection of the two regression lines agrees with the observation that CRABS generally is (up to 5 times) faster than baseline for notebooks with fewer than 14 ambiguities, and (as much as 7 times) slower than baseline when there are more ambiguities than this.  Regression versus total number of information flows in each notebook is more predictive of baseline cost with an R-squared of 0.76. For the 4 notebooks with no ambiguities to be resolved (i.e. for each cell in these notebooks the parser-generated lower and upper estimates of the inter-cell I/O sets are identical), CRABS is 3 orders of magnitude faster than baseline (because no prompting of the LLM is necessary); these 4 notebooks were excluded from the linear regressions just mentioned. We will add to the appendix the two scatter plots (latency vs ambiguities and latency vs information flows) with the regression lines for the two methods on each.
> > >
> > > The above analysis is for the straightforward implementation of CRABS described in the paper which makes each request to LLM sequentially, blocks on each response to each request, and resolves exactly one ambiguity per request.  We subsequently have found that if these requests are made concurrently, CRABS performs between 4 and 24 times faster than baseline on a per-notebook basis. The resulting per-notebooks latencies range from 0.6 to 8.3 seconds with this implementation, which is somewhat closer to what is needed for a practical, interactive application. We are investigating other performance optimizations including packing all questions about a particular cell into a single prompt.

---

### Official Review · Reviewer_rU2s · 2025-05-12

**Rating:** 7
**Confidence:** 4
**Ethics Flag:** 1

**Summary:**

The paper proposes a notebook understanding strategy that models information flow and transitive dependencies. The paper motivates the problem by stating an interesting challenge of understanding api calls in Python notebooks, specifically invocations that may or may not change the state of variables and data contained within the notebook, for which the code is not present in the notebook but rather defined in a library that is imported. The technique relies on leveraging LLM to understand which variables are being mutated without actually executing the code within the notebook. The proposed technique (CRABS) sets out to under-approximate and over-approximate the input and outputs to each Python notebook data cell and then uses an LLM to resolve the ambiguities between the parsing strategies. Finally the paper emperically shows the the CRABS strategy can improve the performance of gpt-4-o. The authors also manually provide a dataset with 50 annotated notebooks for the evaluation set.

**Questions To Authors:**

1. Could you please provide insights into the other cases that fail (apart from the 4 where the MinIOParser and MaxIOParser results match) where the model fails given that even though the LLM resolves 98% ambiguities correctly, the Exact match number over the dataset is only 74% ? Does that mean that there are inherent limitations in the MinIOParser / MaxIOParser -- that model the information flow graph wrong/differently despite the LLM performing well on disambiguating ?

Please refer to the weaknesses.

**Reasons To Accept:**

1. The authors motivate the problem well.
2. The authors empirically validate the long context challenge in understanding notebooks.
3. The information flow analysis for Python notebooks is an interesting approach.

**Reasons To Reject:**

1. There is a strong assumption that global variables cannot be used in the notebooks.
2. Further, there is an assumption that a given notebook only connects a single given workflow, whereas in real situations, there might be multiple workflows that a given user might need to perform that would contain numerous disjoint graphs.
3. The hidden modification of datasets, as pointed out by the authors, is a severe limitation of the technique as they do not truly capture information flow, but rather only the assignment flow, and hence, reasoning, the technique can only reason in cases where there are direct assignments. This makes the technique brittle and might cause hallucinations in the LLM output as well.
4. Also the authors claim that static analysis tools might not recognise output variables in conditional statements, but do not provide evidence that the LLM can successfully (based on the MinIOParser and MaxIOParser) make informed decisions during such cases.
5. It would be interesting to test this strategy not just on GPT-4-o but also other open (Qwen, Llama) as well as closed source(Gemini, Claude) general and code LM (QwenCoder, CodeLlama) to validate the the technique generalises.

---

> ### Author Response · Authors · 2025-06-03
>
> Thank you for your valuable feedback! We address your comments below. Please let us know if you have any follow-up questions.
>
> For **W1**, **W2**, and **W3**, please check the "clarification of the role of syntactic parsers" section in the general responses first.
>
> **Answer to W1**: With respect to global variables, the specific limitation of the current implementation of CRABS is that it does not identify information flows that arise from accesses or updates of global variables within function calls, i.e. flows via variables not mentioned explicitly at the function and method call sites. We fully agree that a practical implementation of CRABS must handle flows arising from global variable accesses (global variables are commonly employed to implement notebook-wide constants) and global variable updates (which may be considered poor practice in some communities but are still, for this reason, very important to handle in a real-world implementation) within functions.
>
> **Answer to W2**: Our current implementation can in fact handle notebooks with multiple workflows. We exclude these notebooks from our dataset because such notebooks are not the focus of our work.
>
> **Answer to W3**: Our current implementation detects hidden modifications by direct assignment (e.g., x=y) or by inclusion in collections (i.e., mutable objects stored in a list, a tuple, or a dictionary). This covers all instances of shared references in our dataset; we will make this clear in the next version of the paper.
>
> **Answer to W4**: Take the same case (Figure 14) using the CRABS input prompt template (Figure 16) as an example. Given the following prompt, the LLM returns: No, “count” is not an input.
> ```
> Given a Python program block, determine if an object is an input. An input is a variable that is used in the program block but not defined within it, or variables used before being reassigned.
>
> Important Cases:
> 1. Conditional Statement Within a Loop A variable may be initialized during an early iteration of a loop and then utilized in subsequent iterations. In such cases, the variable is not considered an input, as its value originates from the loop's execution rather than external sources.
>
> Question:
> In the following Python program block, is "count" an input? Return the final answer in the form "Yes(No), "{x}" is (not) an input" where "{x}" is the actual name of the variable.
>
> for i in range(3):
>     if i ==0:
>         count = 0
>     else:
>         count = count + 1
>
> Answer:
> ```
> **Answer to W5**: Due to time and cost limitations, we tested GPT-4o-mini, Qwen3-8B, and Qwen2.5-Coder-7B-Instruct models. Table R1 (under Reviewer dByL) validates the effectiveness of CRABS over smaller, open-source, general and code language models.
>
> **Answer to Q1**: The low value (74%) of exact match (EM) is reasonable given its strict definition, i.e., the percentage of notebooks for which the entire information flow graph for the notebook (i.e. every one of the individual information flows in that notebook) exactly matches the ground truth. Take the notebook ID#27 as an example. It has 41 code cells and 286 information flows. An LLM correctly resolved 59 of 60 ambiguities that remain after analyzing the syntactic structure of this notebook. However, the single incorrect answer for one ambiguity leads to at least one wrong information flow, disqualifying this notebook from being included in the EM numerator. According to our results, all of the questions in each of 37 notebooks (74%) were answered correctly. Only a few questions (sometimes just one) were answered incorrectly in the remaining 13 notebooks (26%).

---

> > ### Comment · Reviewer_rU2s · 2025-06-04
> >
> > I thank the authors for the detailed response. I hope the authors can make a more general version of CRABS using a rigorously validated syntactic processor. I was looking at the supplimentary material and just realised that the `data/inputs` is an empty folder. Please could the authors rectify that?

---

> > > ### Author Response · Authors · 2025-06-05
> > >
> > > Thank you for identifying the empty `data/inputs` folder. This is *intentional* due to copyright restrictions--we cannot redistribute the Kaggle notebooks directly as they are subject to individual author licenses. This folder serves as a placeholder within our repository structure, designating the location where notebooks extracted from the [Meta Kaggle Code](https://www.kaggle.com/datasets/kaggle/meta-kaggle-code/data) dataset are stored and subsequently accessed when repeating our experiments.
> > >
> > > **To reproduce our experiments**: All required notebooks are publicly available through the Meta Kaggle Code dataset. Please follow the *Data preparation* section in our supplementary material's readme file. We will add one more readme file to the empty `data/inputs` folder explaining the copyright situation and setup instructions.
> > >
> > > **For quick inspection**: If you'd like to examine specific notebooks without downloading the full dataset, here are direct links:
> > > | ID | Notebook URL |
> > > |----|-------------------|
> > > | 01 | https://www.kaggle.com/code/alexisbcook/titanic-tutorial
> > > | 02 | https://www.kaggle.com/code/pmarcelino/comprehensive-data-exploration-with-python
> > > | 03 | https://www.kaggle.com/code/startupsci/titanic-data-science-solutions
> > > | 04 | https://www.kaggle.com/code/serigne/stacked-regressions-top-4-on-leaderboard
> > > | 05 | https://www.kaggle.com/code/yassineghouzam/introduction-to-cnn-keras-0-997-top-6
> > > | 06 | https://www.kaggle.com/code/ldfreeman3/a-data-science-framework-to-achieve-99-accuracy
> > > | 07 | https://www.kaggle.com/code/janiobachmann/credit-fraud-dealing-with-imbalanced-datasets
> > > | 08 | https://www.kaggle.com/code/willkoehrsen/start-here-a-gentle-introduction
> > > | 09 | https://www.kaggle.com/code/tanulsingh077/deep-learning-for-nlp-zero-to-transformers-bert
> > > | 10 | https://www.kaggle.com/code/ash316/eda-to-prediction-dietanic
> > > | 11 | https://www.kaggle.com/code/yassineghouzam/titanic-top-4-with-ensemble-modeling
> > > | 12 | https://www.kaggle.com/code/dansbecker/your-first-machine-learning-model
> > > | 13 | https://www.kaggle.com/code/ybifoundation/stars-classification
> > > | 14 | https://www.kaggle.com/code/dansbecker/model-validation
> > > | 15 | https://www.kaggle.com/code/abhishek/approaching-almost-any-nlp-problem-on-kaggle
> > > | 16 | https://www.kaggle.com/code/dansbecker/underfitting-and-overfitting
> > > | 17 | https://www.kaggle.com/code/gunesevitan/titanic-advanced-feature-engineering-tutorial
> > > | 18 | https://www.kaggle.com/code/karnikakapoor/customer-segmentation-clustering
> > > | 19 | https://www.kaggle.com/code/faressayah/stock-market-analysis-prediction-using-lstm
> > > | 20 | https://www.kaggle.com/code/benhamner/python-data-visualizations
> > > | 21 | https://www.kaggle.com/code/kanncaa1/feature-selection-and-data-visualization
> > > | 22 | https://www.kaggle.com/code/jagangupta/time-series-basics-exploring-traditional-ts
> > > | 23 | https://www.kaggle.com/code/gusthema/house-prices-prediction-using-tfdf
> > > | 24 | https://www.kaggle.com/code/apapiu/regularized-linear-models
> > > | 25 | https://www.kaggle.com/code/dansbecker/random-forests
> > > | 26 | https://www.kaggle.com/code/dansbecker/submitting-from-a-kernel
> > > | 27 | https://www.kaggle.com/code/nadintamer/titanic-survival-predictions-beginner
> > > | 28 | https://www.kaggle.com/code/ybifoundation/simple-linear-regression
> > > | 29 | https://www.kaggle.com/code/kanncaa1/plotly-tutorial-for-beginners
> > > | 30 | https://www.kaggle.com/code/rtatman/data-cleaning-challenge-handling-missing-values
> > > | 31 | https://www.kaggle.com/code/residentmario/indexing-selecting-assigning
> > > | 32 | https://www.kaggle.com/code/ybifoundation/fish-weight-prediction
> > > | 33 | https://www.kaggle.com/code/ybifoundation/cancer-predictio
> > > | 34 | https://www.kaggle.com/code/ybifoundation/chance-of-admission
> > > | 35 | https://www.kaggle.com/code/ybifoundation/multiple-linear-regression
> > > | 36 | https://www.kaggle.com/code/omarelgabry/a-journey-through-titanic
> > > | 37 | https://www.kaggle.com/code/ybifoundation/purchase-prediction-micronumerosity
> > > | 38 | https://www.kaggle.com/code/dansbecker/handling-missing-values
> > > | 39 | https://www.kaggle.com/code/ybifoundation/credit-card-default
> > > | 40 | https://www.kaggle.com/code/ybifoundation/tutorial-decision-tree-regression-scikit-learn
> > > | 41 | https://www.kaggle.com/code/deffro/eda-is-fun
> > > | 42 | https://www.kaggle.com/code/gpreda/santander-eda-and-prediction
> > > | 43 | https://www.kaggle.com/code/kashnitsky/topic-1-exploratory-data-analysis-with-pandas
> > > | 44 | https://www.kaggle.com/code/kushal1996/customer-segmentation-k-means-analysis
> > > | 45 | https://www.kaggle.com/code/kenjee/titanic-project-example
> > > | 46 | https://www.kaggle.com/code/christofhenkel/how-to-preprocessing-when-using-embeddings
> > > | 47 | https://www.kaggle.com/code/sinakhorami/titanic-best-working-classifier
> > > | 48 | https://www.kaggle.com/code/shrutimechlearn/step-by-step-diabetes-classification
> > > | 49 | https://www.kaggle.com/code/ybifoundation/binary-classification
> > > | 50 | https://www.kaggle.com/code/ybifoundation/multi-category-classification

---

> > > > ### Comment · Reviewer_rU2s · 2025-06-05
> > > >
> > > > Thank you for your reply. I have increased my score

---

### Official Review · Reviewer_dByL · 2025-05-21

**Rating:** 6
**Confidence:** 3
**Ethics Flag:** 1

**Summary:**

This paper introduces a new approach, CRABS (Capture and Resolve Assisted Bounding Strategy), for understanding the intricate information flows and operational dependencies within Python notebooks. It proposes a hybrid syntactic-semantic methodology that combines Abstract Syntax Tree (AST) parsing with Large Language Models (LLMs) to construct information flow graphs and cell execution dependency graphs. The paper also contributes an annotated benchmark dataset, comprising 50 highly-upvoted Kaggle notebooks, for evaluating notebook understanding.

**Questions To Authors:**

1. Given the high performance on the benchmark, I am curious whether advanced reasoning models like OpenAI o1/o3,  Gemini-Pro, and Deepseek-R1 could solve this completely without syntactic prompting.
2. There are existing code execution (understanding) benchmarks like CRUX-Eval (Gu, 2024) and Code I/O (Li, 2025). While not identical, they share some common spirits. Could a comparison with these in the related works section be considered?

**Reasons To Accept:**

- The proposed CRABS method effectively reduces hallucination in notebook understanding.
- The creation and annotation of a dataset of 50 real-world Kaggle notebooks provide a valuable resource for future research in notebook understanding and LLM application in code analysis.
- The writing is pretty clear and well-organized, offering rich context and technical details.

**Reasons To Reject:**

1. **Lack of Problem Significance and Practicality:** While the paper articulates the challenges of re-execution due to data and software dependencies, the fundamental importance of deriving information flow *without execution* could be more thoroughly substantiated. For example, in many practical scenarios, especially during development or debugging, direct execution or a robust execution environment might offer a more straightforward and reliable approach for understanding notebook behavior. The paper's argument for avoiding re-execution, while valid, may not universally outweigh the benefits of direct execution for comprehensive analysis. While safety concerns could strongly favor static analysis over execution, this aspect is not prominently highlighted in the Kaggle-based benchmark. Furthermore, the very high success rate (98% F1) of the current method suggests the benchmark might not be challenging enough for evaluating new LLMs. It would be beneficial to include more challenging and compelling examples, both in the benchmark and within the paper's discussion, to better demonstrate the problem's significance.

2. **Lack of Novelty in The Method**: The core method, which combines AST (min and max I/O) parsers and LLMs, while effective, represents a common practice in prompt engineering, leveraging symbolic tools alongside neural LLMs, e.g., Dater (Ye et al., 2023), Binder (Cheng et al, 2023), Maieutic Prompting (Jung 2022). The approach, though functional, might be considered a straightforward application of current capabilities rather than a significant leap in technological contribution.

---

> ### Author Response · Authors · 2025-06-03
>
> Thank you for your valuable feedback! We address your comments below. Please let us know if you have any follow-up questions.
>
> **Answer to W1**: (1) Importance of notebook understanding **without re-execution**. We acknowledge that executing a notebook cell-by-cell to understand its function is common practice and can be highly effective in particular settings, e.g., when authoring and debugging one’s own notebooks or when adapting others’ notebooks for your own purposes. On the other hand, running notebooks that one discovers on various online resources and wish to evaluate as candidates for reuse or repurposing can require significant time and computational resources, especially if the dataset associated with the notebook is large or the notebook has complex software dependencies (as is often the case in scientific applications). So while we obtained the notebooks for our dataset from Kaggle, we expect that solutions to the problems we address will have broader applicability. Pimentel et al. (2019) studied 1.4 million GitHub notebooks and found that only 24.11% could be executed without errors, highlighting the value of avoiding re-execution. We will add these details to the introduction section of our paper.
> (2) The benchmark is still **challenging** for smaller, cheaper, and faster models that would be more suitable for practical applications of CRABS and that we plan to employ as part of our larger research agenda. Conducting the same experiments using GPT-4o-mini, Qwen3-8B, and Qwen2.5-Coder-7B-Instruct models achieves F1 scores of 83%, 75%, and 77% for identifying information flows respectively. Detailed results can be found in Table R1 which will be added to the appendix. In addition, we expect that this benchmark will serve us well as we employ the broad range of Qwen 3 model sizes in our planned efforts to **identify the information and skills** employed by LLMs to solve the various sub-problems comprising the overall notebook-understanding task.
>
> **Answer to W2**: While combining symbolic tools and LLMs is a common practice in prompt engineering generally, researchers have  explored diverse approaches and  frameworks for doing this. Dater (Ye et al., 2023) leverages LLMs to decompose a task into logical sub-questions, where each sub-question can be answered independently using an SQL query; Binder (Cheng et al., 2023) maps a question into a program; and Maieutic Prompting (Jung et al., 2022) generates a maieutic tree using an LLM and employs MAX-SAT to infer the True/False answer. Among them, Dater (Ye et al., 2023) is the most similar to our work. But while Dater uses an LLM to decompose the task and a SQL query to solve a sub-task, CRABS uses syntactic parsers to decompose the task and an LLM to solve a sub-task. Furthermore, our CRABS is a pincer strategy that places bounds both on the total workload assigned to the LLM, and on the individual responses that it can return.
>
> **Answer to Q1**: We have tested the baseline approach (see Figure 15 in the Appendix for an example prompt) using OpenAI o3 with high reasoning mode. It achieved an average F1 score of 94% identifying information flows across our 50 notebooks. We observe that the LLM still fails to understand one of these notebooks, returning the wrong number of cells and earning zero values for all metrics. Furthermore, one of these notebooks (ID#15) takes around 15 minutes to get the answer—about 10 times longer using CRABS and GPT-4o.
>
> **Answer to Q2**: Thank you for referring us to the works related to code execution. CRUXEval (Gu et al., 2024) provides examples of successes and failures of GPT-4 for input and output predictions of Python functions. CodeI/O (Li et al., 2025) trains LLMs on code input-output predictions to improve their general reasoning ability. While LLMs may employ similar skills to solve these various tasks, our focus is information flows and the CRABS strategy for identifying them. We currently are investigating what information is used and what skills an LLM employs to solve such tasks, and the papers you cite will provide a valuable context for that work.  We agree it would be extremely interesting to compare the information and skills employed by these different code understanding tasks.

---

> > ### Author Response · Authors · 2025-06-03
> >
> > | LLM | Method | Prec.(%) | Rec.(%)| F1(%) | Acc.(%) | EM(%) |
> > |-------|----------|------------|----------|--------|----------|----------|
> > | GPT-4o-mini | Baseline | 43.19 | 40.65 | 41.53 | 37.40 | 12 |
> > |  | CRABS |**83.01** | **82.87** | **82.94** | **74.85** | **38** |
> > |  | w/o syntactic phase|71.58 | 65.16 | 67.73 | 55.09 | 10 |
> > |  | w/o cell-by-cell prompting| 54.69 | 53.57 | 54.01 | 50.55 | 28 |
> > | Qwen3-8B | Baseline |46.69 | 39.71 | 42.21 | 35.48 | 6 |
> > |  | CRABS |75.35 | **74.03** | **74.60** | **63.90** | **22** |
> > |  | w/o syntactic phase|**87.42** | 45.13 | 56.65 | 43.17 | 8 |
> > |  | w/o cell-by-cell prompting|37.83 | 37.61 | 37.70 | 32.95 | 16 |
> > | Qwen2.5-Coder-7B-Instruct | Baseline |19.47 | 13.87 | 15.80 | 12.64 | 2 |
> > |  | CRABS |**77.13** | **76.01** | **76.52** | **66.21** | **26** |
> > |  | w/o syntactic phase| 64.90 | 56.64 | 59.81 | 47.24 | 6 |
> > |  | w/o cell-by-cell prompting|35.55 | 33.49 | 34.36 | 31.71 | 16 |
> >
> > Table R1: Information flows: results for different LLMs.

---

> > > ### Comment · Reviewer_dByL · 2025-06-07
> > > **Reviewer Response**
> > >
> > > Thank you for the thorough response. You have convinced me of the problem's significance by demonstrating how it can be leveraged to auto-validate code data quality and executability. Extracting highly usable code can be helpful for either developers' use or LLM training. I believe this problem is more general than notebook static analysis and extends to code analysis with LLMs, potentially at a repository level.
> > >
> > > While I am now sufficiently motivated by the problem, I still have concerns about the practicality of the benchmark given the extremely high accuracy achieved by frontier reasoning models like o3. For research purposes, open-source models already have a very wide range of evaluation benchmarks available, and adding one specialized benchmark may not be a priority for general model evaluation, despite its importance in this crucial field.
> > >
> > > In summary, this is now a good paper from my perspective overall, and I will increase my score to 6, leaning toward seeing it at COLM. Thanks.

---

> > > > ### Author Response · Authors · 2025-06-09
> > > >
> > > > Thank you very much for the encouraging comments with respect to the significance of the problem we are seeking to address.  We agree it is probably better to view the CRABS method and dataset as representative of a specific set of code-related challenges, rather than as a general means of assessing the code reasoning capabilities of LLMs. As you say, there are well-established benchmarks for this.  At the same time, we expect that the general code execution and understanding tasks and benchmarks you refer to will provide a valuable context for future investigations into what information is used and what skills are employed by LLMs  asked to detect information flows in notebooks, and in particular how such necessary information and skills vary across code-related tasks.

---

### Author Response · Authors · 2025-06-03
**General Responses**

We thank all of the reviewers for their very insightful criticisms, questions, comments, and suggestions. We believe that the following general responses provide details missing in the submitted version of the paper that, in light of reviewer feedback, may be helpful for understanding both our intentions in performing the research and our interpretation of the results. We propose to add similar details (in terser form) to the next revision of the paper.  These details fall into three categories: motivation and justification for the specific data set we prepared and worked with; the specific role intended for the implementation of the syntactic parsers employed here; and the overall research agenda in which this work is situated. The latter details will appear in a new Future Work section at the end of the next version of the paper. We hope that together the responses here better contextualize our approach to the research.

**Clarification of notebook selection**

The hypothesis that we intended to test with our work and dataset was whether an LLM could--as an experienced human data scientist can--reason about the information flows in a notebook in cases where this reasoning necessarily must go beyond straightforward parsing of the code in the cells. From our own experience as software engineers and data scientists we know that this task is easier (for humans) when the notebook is in some sense "high quality" and consequently more likely to be reusable and repurposable. We selected highly up-voted Kaggle notebooks for testing this particular hypothesis under the assumption that the degree of up-voting is (in part) an indication of greater relative clarity, reusability, and repurposability of the notebooks. In short, we wanted to focus on notebooks that likely would be clear to humans despite numerous (syntactic) ambiguities in the dataflow to test whether an AI could perform similarly under these ideal conditions. As reviewer(s) point out, this is not an appropriate data set for testing other hypotheses, e.g. whether an LLM can reliably identity the information flows in "average" or "low-" quality notebooks with as many (or more) syntactic ambiguities as these high-quality notebooks exhibit. It is worth noting that even these presumably high-quality notebooks have on average roughly 30 cell I/O ambiguities each. Going forward we fully intend to test CRABS on notebooks more representative of those that are not intended for external review, reuse, etc or are not yet complete, e.g. to facilitate development and debugging of notebooks in the first place. Very different data sets likely will be needed for this future research.

**Clarification of the role of the syntactic parsers**

The primary purpose of the syntactic parsers described in the paper is to validate the effectiveness of the CRABS strategy in the context of the data set under study. We realize that the limitations of their implementation make them unsuitable for a general-purpose tool applicable to all Python notebooks. The syntactic parsers in a sense are a “stand-in” for the more complete, rigorously validated syntactic-analyzer that would be required to deliver a robust, generally useable tool, but they nonetheless correctly perform their role–of producing lower and upper estimates of inter-cell I/O sets that bound the ground truth information flow graph of a notebook–for the 50 notebooks in our data set. We believe that the current implementation is sufficient to support our claim that the pincer strategy by which CRABS performs the notebook understanding task is effective on the notebooks comprising our dataset. On the basis of these results we plan to implement a general-purpose version of CRABS that incorporates a more comprehensive and rigorously validated syntactic processor.

**General research agenda**

This paper represents a first report of a larger effort not only to evaluate LLM capabilities for analyzing and making transparent the function and mechanisms of data science (and scientific) notebooks, but also–and more importantly–to more broadly understand what precisely is involved–the information used, the skills employed, and the sub-problems that must be solved– for any intelligence, artificial or natural, to understand a Python notebook via information flow analysis. We view CRABS as the foundation for an experimental platform for investigating these and related questions. Answering these questions will require different datasets and different statistical analyses than those employed here.

---

### Decision · Program_Chairs · 2025-07-08

**Decision:**

Accept

**Comment:**

Reviewers generally view the proposed CRABS method positively. Some of the highlighted strengths include: a well-motivated problem with an innovative approach combining syntactic parsing and LLM, demonstrating significant performance improvements and providing a valuable resource for computational understanding of non-executed notebooks.

While the reviewers also point out some limitations of this work, they are satisfied with the detailed responses from the authors, which provide more explanations and additional results. Including them in the paper will no doubt improve the paper's quality further, and thus is highly encouraged.